# Plant Extracellular Vesicles with Complex Molecular Cargo: A Cross-Kingdom Conduit for MicroRNA-Directed RNA Silencing

**DOI:** 10.3390/genes17010052

**Published:** 2026-01-01

**Authors:** Ashmeeta Shalvina, Nicholas A. Paul, Scott F. Cummins, Andrew L. Eamens

**Affiliations:** 1Seaweed Research Group, University of the Sunshine Coast, Maroochydore, QLD 4558, Australia; ashmeeta.shalvina@research.usc.edu.au (A.S.); npaul@usc.edu.au (N.A.P.); scummins@usc.edu.au (S.F.C.); 2Centre for Bioinnovation, University of the Sunshine Coast, Maroochydore, QLD 4558, Australia; 3School of Science, Technology and Engineering, University of the Sunshine Coast, Maroochydore, QLD 4558, Australia; 4School of Health, University of the Sunshine Coast, Maroochydore, QLD 4558, Australia

**Keywords:** plant extracellular vesicle (PEV), microRNA (miRNA), gene expression regulation, plant development, environmental stress, pathogen defence, artificial miRNA (amiRNA) technology, biotechnological applications, algae, seaweeds

## Abstract

Plants secrete a heterogenous population of membrane-enclosed extracellular vesicles that harbour an incredible diversity of molecular cargo. It is the complexity of the molecular cargo encapsulated by plant extracellular vesicles (PEVs) which facilitates the fundamental role PEVs play in mediating communication and signalling. PEV molecular cargo is composed of a diverse mixture of lipids, metabolites, proteins, and nucleic acids. Among the nucleic acids, the microRNA (miRNA) class of small regulatory RNA can be viewed as one of the most biologically relevant. Plant miRNAs regulate the expression of genes essential for all aspects of development as well as to control the gene expression changes required to drive the adaptive and defensive responses of plants to environmental stress and pathogen attack. Furthermore, recent research has shown that specific miRNA cohorts are selectively packaged into PEVs as part of the molecular-level response of a plant to its growth environment. For example, PEVs are loaded with a specific miRNA population for their targeted delivery to sites of pathogen infection in the host plant, or for cross-kingdom delivery of host-plant-encoded miRNAs to the pathogen itself. Here we outline PEV physical properties, compare PEV biogenesis pathways, detail the composition of PEV molecular cargo, and go on to provide detailed commentary on the role of PEV-delivered miRNAs in plant development, environmental stress adaptation, and pathogen defence. We conclude this article with a proposal for the potential future use of PEVs and their miRNA cargo in agriculture and aquaculture.

## 1. The Discovery of Plant Extracellular Vesicles

Plant extracellular vesicles (PEVs) are a heterogeneous group of biovesicles enclosed by a lipid bilayer membrane and which range in size from nanometres (nm) through to micrometres (μm) [1,2]. PEVs harbour an extensive diversity of molecular cargo, and it is the complexity of this cargo which facilitates the fundamental role performed by these encapsulated particles in mediating communication and signalling. PEV-mediated communication and signalling occurs both within the plant itself (root-to-shoot signalling, for example) and externally with other organisms in cross-kingdom communication [2,3,4]. In plants, extracellular vesicles (EVs) were first observed in 1965 in cotton (*Gossypium* sp.) synergids (specialised cells which assist in pollen tube guidance to the egg cell in the female gametophyte), being described as single-membraned spheres, and with similar observations soon following in carrot (*Daucus carota* subsp. *sativus*) and wheat (*Triticum* sp.) [5,6]. However, as for the initial discoveries of EVs in other eukaryotes, such as those recorded in animal tissues and fluids, PEVs were initially thought to be mere biological artifacts, likely forming a cellular debris intermediate of either organelle or cell breakdown [3,5]. However, the subsequent identification of EVs in lower eukaryotes and prokaryotes, in addition to their documentation in both plant and animal systems, directly challenged this initial theorem, leading to the conclusion that EVs were indeed biologically relevant entities [7]. Moreover, the repeated reporting of EVs and EV-like particles across highly diverse taxa strongly suggests evolutionary conservation of this particle class given their core importance in directing communication and signalling as part of cellular life [1,8,9].

The transition from the initial view of PEVs simply being cellular byproducts to their recognition as a fundamentally important conduit for cell-to-cell and cross-kingdom communication occurred in the early 2000s when transmission electron microscopy (TEM) was used to study plant–pathogen interactions at sites of infection [10,11]. Localisation of PEVs to pathogen infection sites catalysed a new wave of scientific research aimed at uncovering the full extent to which PEVs, or the EVs from the pathogen, directed molecular communication between the host plant and invading pathogen [12,13,14]. This research also showed that in addition to supplying molecular cargo for pathogen defence, these encapsulated particles ferry specific molecules as part of normal development or to enable a plant to respond to environmental stress [15,16,17]. Further research focus was additionally placed on the cross-kingdom communication role directed by PEVs following the Zhang et al. [18] study. Specifically, Ref. [18] reported members of highly conserved plant microRNA (miRNA) families, including members of the miR156, miR166, and miR168 families, in human and mouse digestive fluids and sera post-ingestion of diets rich in rice (*Oryza sativa*) and brassicas such as Chinese cabbage (*Brassica rapa pekinensis*). Moreover, the abundance of these dietary plant miRNAs was enriched in circulating EVs, which implied selective packaging of these plant miRNAs into mammalian EVs [18]. This finding also indicated that dietary-consumed plant miRNAs would have initially been packaged into PEVs to survive the harsh environment of the mammalian gastrointestinal tract, prior to transfer of the packaged miRNA cargo into mammalian EVs [19]. Indeed, today, PEVs harbouring molecular regulatory molecules, such as miRNAs, have been isolated from homogenised tissues or juices extracted both from edible plants (including barley (*Hordeum vulgare*), broccoli (*Brassica oleracea* var. *italica*), carrot, cabbage (*Brassica oleracea* var. *capitata*), garlic (*Allium sativum*), ginger (*Zingiber officinale*), grape (*Vitis vinifera*), lemon (*Citrus limon*), orange (*Citrus* × *sinensis*), potato (*Solanum tuberosum*), rice, strawberry (*Fragaria* × *ananassa*), and wheat) and from medicinal plants (such as *Houttuynia cordata* (fish mint), *Momordica charantia* (bitter lemon), *Morus alba* (white mulberry), *Morus nigra* (black mulberry), *Panax notoginseng* (ginseng), and *Solanum nigrum* (black nightshade)) [20,21,22,23,24,25].

Among the diverse molecular cargo of PEVs, the miRNAs, a predominant species of small regulatory RNA (sRNA) which accumulate in the cells of all eukaryotes [26,27,28] and which regulate target gene expression at the posttranscriptional level [26,27,28], can be viewed as one of the most biologically relevant bioactive molecules. In plants, miRNAs direct expression regulation of target genes which display a high degree of sequence complementarity to the targeting miRNA via this sRNA class directing a messenger RNA (mRNA) cleavage mode of RNA silencing [29,30,31]. This form of miRNA-directed target gene expression regulation is crucial to all aspects of plant development, acting both locally by defining the expression boundary of developmentally important genes at specific stages of development [32,33] and globally via controlling the transition from vegetative-to-reproductive development [34,35]. miRNA-directed target gene expression regulation is also central to a plant to mount an adaptive response to environmental stress [36,37,38] or to defend against plant pathogens, including viral, fungal, and bacterial pathogens [39,40,41]. Here, we outline the physical properties of PEVs, compare PEV biogenesis pathways, detail the molecular cargo of PEVs, and go on to provide detailed commentary on the role of PEV-delivered miRNAs in plant development, environmental stress adaptation, and pathogen defence. We conclude this article with a proposal for the potential future use of PEVs and their miRNA cargo in the agricultural setting.

## 2. Physical Properties of Plant Extracellular Vesicles

Advanced microscopy techniques such as cryogenic electron microscopy (cryo-EM) and TEM have shown PEVs to exhibit considerable variability in their physical dimensions [1,5,6]. This is thought to stem from a combination of factors, including the following: (1) the specific biogenesis pathway, (2) the source material under analysis (e.g., plant species and/or specific tissue of origin), (3) the cultivation and/or environmental conditions at the time of sampling (e.g., normal growth conditions or exposure to environmental stress), and (4) the extraction protocol employed for PEV isolation/purification [1,8,42]. Cryo-EM studies have shown that PEVs have an average diameter of approximately 110 nm, but considering that PEV isolations are complex biovesicle mixtures, a general particle size range of 30 to 500 nm is commonly reported [1,6]. Assessment of PEV isolations via other approaches suggests greater variation, with PEV diameter ranges of 50 to 1500 nm (1.5 μm) reported [5,6,8]. However, it is important to note that the assessed preparations may also contain “contaminants”, such as intracellular vesicles, apoplastic vesicles, and artifactual vesicles, either released by cell rupture during tissue homogenisation or which have formed during the purification process itself.

PEV size variability may simply be the result of the predominant biogenesis pathway of a particular plant species. For example, broccoli PEVs have an average diameter of 18–400 nm [43], orange PEV diameters are 105–396 nm [44], ginger PEV diameters range from 125 to 250 nm [45], and grape PEVs have an average diameter of ~400 nm [46]. In contrast, carrots produce a PEV population with considerably varied sizes of diameters ranging from 100 nm to over 1.0 μm (Figure 1A) [1,8,21]. Further adding to PEV size variation is the difference in diameter between single bilayer vesicles and their double-layered counterparts, with single-layered PEVs generally smaller at ~135 nm in diameter compared to double bilayer PEVs, which exhibit a mean diameter of ~320 nm (Figure 1A) [6]. Overall, however, PEVs generally exhibit a larger size compared to the EVs released from animal cells which typically measure between 30 and 150 nm in diameter (Figure 1B) [1]. Comparison of plant to animal EV size poses the intriguing question of whether increased variability in PEV diameter facilitates their functional requirement in a more diverse array of fundamental biological processes compared to those more well-established roles attendant to animal EVs.

Post-isolation, and likely due to being membranous vesicles, PEVs commonly adopt a spherical structure [1,5]. Further, this spherical structure appears to form post-isolation regardless of whether the PEV is enclosed by single or multiple lipid bilayer membranes [5,6]. Biologically, such a structure would facilitate the flexibility required for PEVs to either be actively or passively passaged through the rigid plant cell wall. It is also recognised, however, that the spherical shape commonly observed for PEVs post-isolation may also be due to the physical stresses placed on vesicles during their extraction from plant tissues in combination with the native structural requirements of PEVs to traverse the plant cell wall [2,42]. The influence that the extraction process employed to obtain a PEV preparation has on altering vesicle shape is best evidenced by reports of a change in PEV structure before and after vesicle extraction. Moreover, via TEM, PEVs have been observed as cup-shaped or saucer-shaped structures in planta [47,48], a specific structure also adopted by animal EVs in vivo [49,50]. However, it is important to note that the converse has also been reported; that is, spherical EVs transform into cup-shaped structures during specific steps of the sample preparation process required for TEM analysis, such as the dehydration series used for sample assessment by this microscopy technique [49,51]. Surface depressions are another structural feature shared between plant and animal EVs. Again, however, it is unknown whether these are sample preparation artifacts or if they form a conserved inherent architectural feature of functional significance [2,42].

Although plant and animal EVs appear structurally similar, the membrane composition of PEVs is distinct from that of animal EV membranes. Specifically, PEV membranes contain a high abundance of phosphatidic acid, galactosyldiacyl-glycerol, phosphatidylethanolamine, monogalactosyl-diacylglycerol, and phosphatidyl-choline [52,53]. This is compared to the composition of animal EV membranes which contain high levels of sphingomyelin, cholesterol, gangliosides, phosphatidyl-choline, phosphatidylinositol, ceramide, phosphatidylethanolamine, and diacylglycerol (Figure 1B) [54,55]. The distinction in membrane composition between plant and animal EVs most likely stems from the differences in the passaging requirements for EVs to exit cells and enter the extracellular spaces in plants and animals, respectively. Most notably, a distinct PEV membrane composition is required to enable these vesicles to traverse the thick, rigid walls of plant cells while keeping their molecular cargo intact [5,56,57]. Due to the unique composition of PEV membranes, post-isolation, these biovesicles possess a negative zeta potential, which is generally greater than −20 millivolts (−20 mV) [1,58]. Zeta potential refers to the electrical charge which forms at the interface (referred to as the “shear plane” or “slipping plane”) between the surface of a molecule and its liquid medium, and with respect to nanoparticles such as PEVs, it is used to predict the long-term stability of the particle. The negative zeta potential displayed by PEVs denotes their remarkable stability and underscores their crucial functional role in extracellular communication [1,58], most likely via facilitating their (1) their passage through the plant cell wall, and (2) subsequent uptake into the cells of other organisms which have marked differences in membrane composition.

## 3. Biogenesis Pathways for Plant Extracellular Vesicle Production

The heterogeneity of PEV populations infers the existence of multiple sophisticated and potentially partially redundant biogenesis pathways for EV production in plants [5,56,59]. Furthermore, a diversity in origin would also account for the observed variability of PEV size, structure, membrane composition, molecular cargo and biological function [60,61,62]. Attempts have been made to broadly categorise PEVs into subgroups, each likely originating from one of the multiple biogenesis pathways for PEV production in plants. These pathways include the well-recognised multivesicular body (MVB) and exocyst-positive organelle (EXPO) biogenesis pathways, together with the more recently proposed vacuolar and autophagy biogenesis routes as “other” or “novel” biogenesis pathways [5,56,57,61].

Exosomes are an EV subtype which first form intracellularly when the endosome membrane invaginates to generate intraluminal vesicles (ILVs) within MVBs. Once matured, MVBs fuse with the cell plasma membrane, leading to the release of ILVs into the extracellular space as exosomes [56,60]. In animals, exosome production via the MVB biogenesis pathway is tightly regulated by the Endosomal Sorting Complexes Required for Transport (ESCRT) machinery, with homologues for most ESCRT proteins now identified in plants [63,64,65]. MVBs were first identified in plant cells in 1967, which together with the more recent identification of plant ESCRT machinery protein homologues, show that an ESCRT-regulated MVB biogenesis pathway also makes a significant (most likely predominant) contribution to EV production in plants [56,60,61]. In the animal MVB biogenesis pathway, four protein complexes, including ESCRT-0, ESCRT-I, ESCRT-II, and ESCRT-III, together with specific sets of accessory proteins, form the ESCRT machinery required for molecular cargo recruitment to endosomes and exosome formation from MVBs [66,67,68,69]. Specifically, ESCRT-0 is responsible for recruiting both the molecular cargo to be packaged into exosomes and the ESCRT-I complex to the endosome membrane. Together, ESCRT-0/ESCRT-I subsequently recruit ESCRT-II to the endosome where ESCRT-I/ESCRT-II distorts the membrane inward to form budding structures [66,67,68,69]. Post-internal bud formation, ESCRT-II recruits ESCRT-III to budding sites where it directs bud scission to generate ILVs within MVBs [66,67,68,69]. An accessory protein, VACUOLAR PROTEIN SORTING4 (VPS4), which functions as a triple-A (contains an AAA+ catalytic module) ATPase, mediates the final step of the MVB pathway: dissociation of all ESCRT machinery from the endosome membrane [70,71]. Although homologues with plant-specific adaptations have been identified for most ESCRT machinery proteins [63,64,65], the precise mechanism by which MVBs are targeted to the plasma membrane and released through the cell wall remains less well characterised in plants compared to the animal system [56,60,61]. Furthermore, the plant-specific ESCRT protein FYVE DOMAIN PROTEIN REQUIRED FOR ENDOSOMAL SORTING1 (FREE1) appears to perform all functions of the animal proteins of the ESCRT-0 complex in addition to regulating processes such as vacuolar protein transport and autophagic degradation, as well as to direct protein sorting as part of MVB formation [72,73]. Figure 2 depicts the MVB biogenesis pathway as the predominant PEV production pathway under normal growth conditions, with the other PEV biogenesis pathways (i.e., EXPO, vacuolar, and autophagy pathways) likely induced to make a more significant contribution to the global PEV population when a plant experiences abiotic or biotic stress.

Exocyst-positive organelles (EXPOs) are a group of double-membraned organelles which adopt a spherical structure intracellularly to closely resemble autophagosomes [5,74,75]. EXPOs are involved in unconventional protein secretion processes via their loading with select cytoplasmic proteins for transport to the cell surface for secretion [5,74,75]. In plants, this process facilitates the delivery of soluble proteins which lack conventional signal peptides to the apoplast [57,74,75]. Like the PEVs generated by the MVB biogenesis pathway (Figure 2B), EXPO pathway-produced PEVs display physical properties identifying them as an exosome subtype. MVB pathway-derived exosomes can be separated from EXPO pathway exosomes by their distinct marker protein profiles. Specifically, MVB pathway-derived exosomes express either a high abundance of the molecular cargo recruitment proteins TETRASPANIN8 (TET8) and TET9 [76,77] or the cell wall protein PENETRATION1 (PEN1). In comparison, EXPO pathway-derived exosomes harbour a high abundance of the exocyst protein subunits EXO70B2, EXO70E2, and EXO70H1 [61,77,78]. Moreover, presentation of these exocyst subunits on the lipid bilayer membrane of EXPO pathway-derived PEVs is thought essential for the transport of this exosome subclass of PEV to the cell membrane for secretion [79,80,81]. Furthermore, via detection of an increased abundance of exosomes marked with the EXO70 proteins in PEV isolations, EXPO pathway-derived PEVs have been proposed to be specifically involved in signalling processes as part of a plant mounting a defence response to pathogen attack [61,80,81]. Similarly, considering that EVs are known to be secreted in plants exposed to environmental stress [4], and that external stressors can cause membrane dysfunction, as well as the accumulation of misfolded proteins or protein aggregates to trigger EV formation [82], EXPO pathway-derived PEVs may well form the predominant PEV subclass produced as part of the adaptive response of a plant to environmental stress (Figure 3A) [4,82].

In plant cells, the vacuole is a multifunctional organelle that mediates several aspects of protein storage, modification, and/or degradation to maintain cellular homeostasis [83,84]. Indeed, the initially formed and smaller-sized vacuoles may originate from MVBs to serve specific cellular functions (e.g., transport of unconventionally secreted proteins to the plasma membrane) prior to fusing with other smaller-sized vacuoles as part of the formation of the large central vacuole [85,86]. It is this link between the MVB pathway and central vacuole formation, together with the proposed role of vacuoles in unconventional protein secretion, which has led to the hypothesis that vacuoles may potentially form a third biogenesis route for PEV production [5,60,61]. Furthermore, the contents of vacuoles can be released outside of the cell membrane in vesicular form during pathogen attack, a strategy which would provide an effective approach to plant cells for the specific targeting of an invading pathogen via the localised delivery of defence molecules (Figure 3B) [85,87,88,89]. It has also been hypothesised that, in plants, autophagy may offer yet another alternate biogenesis route for PEV production [60,87]. Moreover, autophagy forms a highly conserved pathway in plants to deliver unrequired proteins or damaged cellular components to the central vacuole for degradation and/or recycling to provide molecular reserves for the replenishment of cellular resources [86,90]. Due to the potentially damaging nature of some of the molecules generated by autophagy, molecules which subsequently require delivery to the vacuole, these molecules are packaged into autophagosomes, a type of double-membraned vesicle, for their safe passage to the vacuole [91,92]. Currently available information suggests that an autophagosome-based contribution to PEV production could potentially be via either their interaction with the central vacuole as an intermediatory step or via direct fusion of autophagosomes to the plant cell plasma membrane for the external release of their contents (Figure 3C) [60,93]. It is important to note that the contribution of these two hypothesised biogenesis pathways (vacuolar and autophagy biogenesis models) to the global profile of the EV population of a plant is likely heavily dependent on the growth environment. Specifically, environmental stress and pathogen attack can cause dysfunction to the plant cell membrane which in turn can lead to the accumulation of misfolded proteins or the misexpression of proteins with altered function, both of which can trigger vesicle formation [4,82,94,95]. This, together with the demonstration that PEVs are known to be involved in stress responses in plants [4,82], indicates that these alternate biogenesis pathways for PEV production may only be actively utilised when the plant’s growth environment is altered by abiotic or biotic stress.

Post their passage across the plasma membrane, PEVs next face the significant challenge of successfully transversing the rigid walls of plant cells to direct their central functional role in communication and signalling. This challenge is inherent to PEVs due to this unique structural distinction of a plant cell to that of the cells found in animal systems. Specifically, the plant cell wall is composed of an intricate network of cellulose microfibrils, cross-linked with embedded glycans and a matrix of pectin polysaccharides [96,97]. In many plant species, a thicker and even more structurally rigid secondary wall forms adjacent to the primary cell wall due to the added deposition of lignin in the secondary wall [96,98]. The composition of both the primary and secondary cell wall forms a dense network with pore spaces often much smaller than the diameter of the PEVs released across the plasma membrane [87,99,100]. However, PEVs obviously possess remarkable physical properties which facilitate their passage through the plant cell wall so that they can perform their in planta or cross-kingdom communication roles [8]. Both a passive and an active mechanism of passage have been proposed for PEVs to traverse this formidable physical barrier. For passive passage, it has been proposed that the specific physical characteristics of plant cell walls, including their inherent dynamics and viscoelastic properties, direct transient structural rearrangements that facilitate PEV passage (Figure 4A) [87]. Alternatively, it has also been suggested that PEVs themselves may be able to mediate their own morphological modifications to transform from a spherical or cupular shape, into an elongated tubular structure which would more readily facilitate their passage through the narrow pores of the cell wall matrix (Figure 4B) [87]. Active PEV passage through plant cell walls has been hypothesised to be based on the activity of cytoplasmic enzymes that pass through the plasma membrane along with the PEVs, enzymes which would locally modify the composition of the cell wall to provide transient pathways for PEVs to exit plant cells (Figure 4C) [60,87]. Interestingly, it has also been suggested that PEVs may facilitate their own passage through the cell wall. More specifically, components of the PEV proteome provides the enzymatic activity required to modify specific structural components of the plant cell wall (e.g., cellulose, pectins, glycans, or lignin) to reduce its integrity, and thereby, increase pore size to facilitate PEV passage (Figure 4D) [101].

## 4. Plant Extracellular Vesicles Harbour Diverse Bioactive Molecular Cargo

A defining feature of all PEV subclasses is their protein–lipid membrane, which encapsulates the bioactive molecular cargo that they harbour: a fundamental structural architecture conserved with animal EVs. The exact protein composition of the lipid membrane differs somewhat depending on the PEV subclass. For example, the membranes of MVB pathway-derived PEVs harbour a high abundance of proteins TET8, TET9, or PEN1 [76,77,78], whereas the membranes of EXPO pathway-derived PEVs contain the exocyst protein subunits EXO70B2, EXO70E2, and EXO70H1 [79,80,81]. Similarly, although not yet well defined, the membranes of PEVs derived from the hypothetical vacuolar and autophagy biogenesis pathways would be expected to harbour a high abundance of membrane proteins specific to these alternate production pathways. Furthermore, all PEV subclasses would also be expected to contain a similar, or shared, cohort of proteins which would function to facilitate the remodelling of the PEV particle itself, or they would drive the structural alterations of cell walls required for PEVs to deliver their molecular cargo externally. For example, of the plant species for which the composition of the PEV membrane has been profiled, a high abundance (albeit at differing ratios) of the lipids phosphatidic acid (PA), digalactosyldiacylglycerol (DGDG), monogalactosyldiacylglycerol (MGDG), and phosphatidyl-choline (PC), together with a range of phytosterols, has been reported [52,53,102]. Harbouring such lipids is expected as they not only are known to be required for EV biogenesis but also serve numerous other EV-associated functions. Such functions include: (1) controlling intercellular molecular interactions, including defining the molecular cargo to be loaded into vesicles; (2) maintaining vesicle stability; (3) mediating vesicle–cell membrane fusion events, and; (4) directing the cellular release of mature PEVs via exocytosis [45,52,103]. Taking the phospholipid bilayer of ginger and grapefruit (*Citrus* × *paradisi*) as an example, the membrane composition of ginger PEVs contains 25–40% PA, 25–40% DGDG, and 20–30% of MGDG [52]. In contrast, the membranes of grapefruit PEVs only contain 2.5% PA, with the phospholipids phosphatidylethanolamine (~45%) and PC (~30%) predominating [53].

The lipid composition of the internal contents of PEVs from ginger, grape, grapefruit, and lemon was recently reported, with the 465 lipids identified in the PEVs derived from all four plant species analysed, grouped into six categories: (1) fatty acids, (2) glycerolipids, (3) glycerophospholipids, (4) prenol-like alcohols, (5) sphingolipids, and (6) sterol lipids [104]. In addition to this large number of shared lipids, 400, 40, 29, and 87 unique lipids were documented for ginger, grape, grapefruit, and lemon PEVs, respectively [104]. This finding shows that ginger-derived PEVs harbour a lipid cargo profile quite distinct from those of grape, grapefruit, and lemon PEVs. The uniqueness of the ginger PEV lipid profile was likely due to the considerably higher ratio of fatty acids to glycerophospholipids in ginger PEVs compared to grape-, grapefruit-, and lemon-derived PEVs, with the PEVs from these three species all displaying opposing fatty-acid-to-glycerophospholipid ratios. However, like grape and lemon PEVs, ginger PEVs also harbour a high abundance of glycerolipids, while grapefruit-derived PEVs contained a higher glycerophospholipid content [104]. In another study on lipid profiling of PEVs derived from the juice processed from oranges, the content of phosphatidylethanolamine was considerably higher than the content of either PC or PA in these vesicles, and further, the contents of diacylglycerol and other fatty acids such as linolenic, linoleic, oleic, and palmitic acid were much higher in the PEVs isolated from orange juice than in the PEVs from other plant species [44]. A species-specific lipid profile was also reported for the PEVs derived from turmeric (*Curcuma longa*), with phosphatidylethanolamine, triglyceride, phosphatidylinositol, PC, and digalactosylglycerol being the five most abundant lipids in turmeric-derived PEVs [105]. Taken together, such “lipidomic”-focused studies have readily shown that PEVs harbour a diversity of lipids and that the lipid profile is specific to the plant species of PEV origin.

Metabolic profiling of PEVs has shown that in addition to harbouring a diverse lipid profile, these vesicles also carry an extensive range of metabolites [105,106,107]. Again, however, the metabolite composition of PEVs appears to be highly dependent on the plant species of origin. Among the metabolites reported to form PEV cargo are various phenolics, flavonoids, terpenoids, and alkaloids [105,106,107], as well as other active and species-specific compounds such as cannabidiol from *Cannabis sativa* (Cannabis) [108]. For the PEVs characterised for other plant species, a range of additional metabolites which serve important roles such as oxidative stress suppression and the inhibition of phototoxicity, as well as directing anti-inflammatory and anti-cancer effects in humans, have been reported [103,105,109,110]. More specifically, the flavonoid naringin, reported to serve as both an anti-neoplastic and anti-inflammatory agent, is harboured by mandarin (*Citrus reticulata*)-derived PEVs. In addition, the natural antioxidants L-ascorbic acid (Vitamin C) and trans-δ-viniferin are abundant in the PEVs derived from grapefruit and grapes, respectively [107,110]. Furthermore, the phenolic compound anthraquinone, which is commonly used as a laxative, is found in the PEVs derived from various *Aloe* spp. [107], and the anti-inflammatory metabolite curcumin is carried by turmeric-derived PEVs [105]. Finally, in ginger-derived PEVs, the phenolic phytochemical gingerol, which has well-documented anti-inflammatory and anti-nausea properties, is highly concentrated [104].

In the PEVs isolated from the resurrection plant species *Craterostigma plantagineum* (blue carpet), proteomic analysis identified numerous cell-wall-associated proteins including α-L-arabinofuranosidase, β-xylosidase, β-galactosidases, polygalacturonases, 1,3-β-glucosidases, and pectin esterases [53]. In addition to these cell-wall-associated proteins, plasma membrane proteins, such as the aquaporins plasma membrane intrinsic protein 1C (PIP1C) and PIP2-7-like, were also identified in *C. plantagineum* PEVs [53]. Cell-wall-associated proteins and aquaporins were also catalogued among the protein cargo harboured by PEVs derived from ginger, grapefruit, and tomato (*Solanum lycopersicum*) [52,111,112], a finding that identifies these proteins as being preferentially loaded in PEVs. In the PEVs derived from tomato roots, the growth- and development-associated proteins actin, annexin, calmodulin, calreticulin, and H^+^-ATPases with molecular transport roles, as well as various other transporter proteins (e.g., water, nitrate, and phosphate transporters) were highly abundant [112]. Proteomic analysis of PEVs from *Arabidopsis thaliana* (*Arabidopsis*), lemon, grapefruit, ginger, and tomato also repeatedly documented the presence of numerous heat shock proteins (HSPs), such as HSP70, HSP80, and HSP90 [52,111,112]. In plants, HSPs have been associated with numerous functional roles such as maintenance of cell integrity via their involvement in cell cycle regulation, mediating signal transduction processes, and facilitating apoptosis [113,114]. HSPs have also been widely documented to function as chaperone proteins operating with other, and usually closely related, proteins (such as other HSPs) to assist with the folding of their interacting proteins to maintain the function of these proteins when the physiological conditions of a cell are altered, such as during periods of stress exposure [115,116]. Therefore, the repeated identification of HSPs in PEVs identifies a role for PEVs in coordinating a plant’s response to stress.

A recent study [6] used a proteomic approach to compare the protein profiles of the PEVs isolated from the apoplastic wash fluids collected from the monocotyledonous grass *Sorghum bicolor* (sorghum) and the model dicotyledonous species *Arabidopsis*. This analysis revealed that the sorghum EV proteome consisted of 437 proteins, with only 65 (~15%) of the EV-loaded proteins possessing an amino-terminal (N-terminal) signal peptide required for classical protein secretion. This formed an expected finding considering that intracellular and extracellular vesicles are well-established conduits for the transport of unconventionally secreted proteins in both plants and animals [57,74,75]. In contrast, 163 of the 437 (~37%) sorghum EV-specific proteins were determined to house at least a single transmembrane region, a finding which again agreed with previous profiling of the structural characteristics of proteins loaded into plant and animal EVs [117,118]. The authors next assigned the sorghum PEV proteins to functional categories via the use of Gene Ontology (GO) enrichment analysis. When compared to the functional categories of the whole cell proteome of sorghum, GO term enrichment analysis revealed the five most highly enriched categories for sorghum PEV-loaded proteins to be: (1) vesicle docking/fusion; (2) ribosome- and translation-related; (3) transmembrane transport; (4) metabolism (carbohydrate, lipid and amino acid), and; (5) protein folding [6]. The sorghum EV proteome was next compared to the proteome of *Arabidopsis* EVs [76], with this analysis identifying proteins assigned to the same functional groupings in *Arabidopsis* EVs as evidenced by ~60% of sorghum PEV-enriched proteins determined to have a homologue in the *Arabidopsis* PEV proteome [6]. It is important to note here, however, that of the almost 600 proteins detected in *Arabidopsis* PEVs isolated from apoplasmic wash fluid, ~26% were assigned to the two functional categories “biotic stress” and “abiotic stress” following application of the same GO term enrichment approach [76]. Although homologues for some of these *Arabidopsis* PEV harboured stress- and defence-related proteins were also identified in the sorghum PEV proteome [6], the greater percentage observed for the *Arabidopsis* EV proteome [76] suggests that in *Arabidopsis*, PEV-delivered proteins may direct a more central role in stress and defence processes than the proteins packaged into sorghum PEVs.

The PEV lipid bilayer not only protects the integrity of its transported cargo rich in lipids, metabolites, and proteins [4,17,101] but also facilitates the safe passage of an array of nucleic acids. Specifically, the presence of numerous different forms of RNA, including regulatory classes of sRNA, such as the miRNA class of regulatory RNA, is well documented as an abundant PEV molecular cargo [1]. In plants, miRNAs are processed from stem-loop structured precursor transcripts (termed the primary-miRNA (pri-miRNA) and precursor-miRNA (pre-miRNA), respectively) of imperfectly complementary molecules of double-stranded RNA (dsRNA) by the microprocessor complex proteins SERRATE1 (SE1), dsRNA BINDING1 (DRB1), and DICER-LIKE1 (DCL1) [119,120]. In the plant cell nucleus, SE1 together with DRB1 accurately position DCL1 on the pri-miRNA and pre-miRNA precursor transcripts to ensure that just a single miRNA silencing signal is liberated from the considerably longer-length precursors by sequential DCL1-catalysed cleavage of the pri-miRNA and pre-miRNA [119,120]. The resulting miRNA/miRNA* duplex is then loaded into the RNA-induced silencing complex (RISC), which contains the ARGONAUTE1 (AGO1) endonuclease at its catalytic core. AGO1 separates the miRNA guide strand from the miRNA* passenger strand and retains the miRNA guide strand while discarding the miRNA* strand to form miRNA-directed RISC (miRISC). In the plant cell cytoplasm, the now activated miRISC uses the loaded miRNA as a sequence specificity guide to regulate the expression of target genes that possess a miRNA target site sequence with a high degree of complementarity to the sequence of the loaded miRNA. Moreover, in plants, miRISC regulates target gene expression via an AGO1-catalysed mRNA cleavage mode of RNA silencing [119,120]. Plant miRNAs were originally identified as master regulators of developmental gene expression [32,33,34,35] but have since been shown to additionally play a central role in regulating the expression of genes required for a plant to mount an adaptive response to environmental stress [36,37,38] or a defensive response to invading pathogens, including bacterial, fungal, and viral pathogens [39,40,41].

To provide effective protection against invading pathogens, plant miRNAs not only need to modulate host gene expression but also require targeted delivery to the site of pathogen challenge and then subsequent internalisation by the invading pathogen to further control the expression of the pathogen’s virulence-promoting genes. Host plant EVs could potentially facilitate this host-to-pathogen cross-kingdom gene expression regulation role for a plant miRNA. Indeed, Cai et al. [12] provided evidence for such a role via their demonstration that more than 70% of the *Arabidopsis* sRNAs which are transported to the necrotrophic fungal pathogen *Botrytis cinerea* (*B. cinerea*) were present in PEVs. Furthermore, the authors showed that the *Arabidopsis* PEV-enriched sRNA population does not strongly correlate with the abundance of detected sRNA species in host plant leaf cells: the specific organ and tissues responsible for PEV production [3,12]. This finding strongly suggests that *Arabidopsis* sRNAs are selectively loaded into vesicles at the site of PEV production for subsequent transport of the packaged sRNAs to their site of action to enable the tight control of target gene expression. The authors [12] also showed that compared with wild-type *Arabidopsis* plants, considerably fewer sRNAs were transported to sites of *B. cinerea* infection when the fungus was used to infect the *Arabidopsis tet8 tet9* double mutant plant line. As outlined above, the presence of the molecular recruitment proteins TET8 and TET9 are used as markers to identify PEVs produced by the MVB biogenesis pathway [76,77]. Therefore, the demonstration by Cai and colleagues [12] that fewer sRNAs are transported to *B. cinerea* in the *tet8 tet9* double mutant, and that this mutant plant line is more susceptible to *B. cinerea* challenge, strongly implicates the central importance of the transportation role mediated by PEVs to deliver miRNA-directed defence signals to the sites of pathogen attack.

The selective sorting of specific miRNAs into PEVs produced by *Arabidopsis* leaves upon *B. cinerea* challenge also suggested that specific protein machinery would be involved in this process. Accordingly, the proteomic assessment of *Arabidopsis* PEVs post *B. cinerea* infection [118] showed that TET8-positives PEVs harboured a high abundance of RNA binding proteins (RBPs), including AGO1, the primary effector protein of miRNA-directed target gene expression regulation in *Arabidopsis* [119,120]. In addition, ANNEXIN1 (ANN1) and ANN2, RBPs involved in membrane stabilisation and stress responses [121], were also identified. RH11, RH37, and RH52, three RBPs which belong to the DEAD-box RNA helicase (RH) protein family, with these and other members of this protein family shown to be involved in all aspects of RNA synthesis and metabolism [122], were also identified by this RBP-focused assessment [118]. Profiling of the sRNAs bound by each of these RBPs showed that AGO1, RH11, RH37, and RH52 bind to specific sRNA species which are 20 to 22 nt in length and which predominantly contain uracil (U) as the 5′ terminal nucleotide, whereas the ANN1 and ANN2 proteins interacted with PEV-enriched sRNAs less specifically. Together, these findings suggest that the RBPs AGO1, RH11, RH37, and RH52 load specific sRNA-silencing signals into PEVs, while the ANN1 and ANN2 RBPs likely stabilise the packaged sRNAs post-loading [3,118]. As shown for the *Arabidopsis tet8 tet9* double mutant [12], the proposed roles for the PEV-enriched RBPs in miRNA loading and stabilisation in the analysed vesicles were confirmed via the demonstration that the *Arabidopsis* mutant lines *ago1*, *ann1 ann2*, and *rh11 rh37* all had reduced sRNA loading to PEVs (including reduced miRNA abundance) and were more susceptible to *B. cinerea* challenge than were wild-type *Arabidopsis* plants [118]. However, when considered together, analysis of the *ago1* single mutant and the *tet8 tet9*, *ann1 ann2*, and *rh11 rh37* double mutants only revealed PEV-mediated delivery of miRNAs to *B. cinerea* infection sites was reduced, not completely abolished [3,118]. This indicates that other RBPs, sRNA delivery routes, and/or PEV subclasses are also involved in the delivery of miRNA-silencing signals to the sites of pathogen infection.

### 4.1. The MicroRNA Cargo of Plant Extracellular Vesicles and Their Roles in Development and Environmental Stress Adaptation

In *Arabidopsis*, an elegant study [15] catalogued the sRNA population of PEVs from the intercellular spaces of leaves to determine the functional significance of this regulatory class of RNA. Moreover, the authors compared the PEV sRNA population to that of the intercellular wash fluid from which the vesicles had been isolated to identify the sRNA species preferentially loaded into the PEVs generated by *Arabidopsis* leaves [15]. This comparison revealed that of the various sRNA species identified in both the PEV-enriched and intercellular wash fluid (PEV-depleted) samples, including the heterochromatin small-interfering RNA (hcsiRNA), *trans*-acting siRNA (tasiRNA), phased siRNA (phasiRNA), and miRNA species of small regulatory RNA, the miRNAs had the highest degree of differential accumulation. This indicates that in *Arabidopsis* leaves, a miRNA subpopulation is selectively packaged into the PEVs generated by this tissue for their subsequent transportation. This finding also showed that a second, distinct miRNA subpopulation are mobile within *Arabidopsis* leaf tissues; however, the mobility of this subpopulation is independent of a PEV-mediated mode of transport [15]. Of the miRNAs with higher abundance in PEVs than in the EV-depleted intercellular wash fluid were the miR158, miR162, miR394, miR408, miR447, miR869, miR5642, and miR8175 silencing signals. Interestingly, also present at a higher abundance in the PEV-enriched sample were the passenger strands of several miRNA duplexes, sRNAs which form after the separation of the two duplex strands by miRISC, including miR158*, miR167*, miR168*, miR390*, and miR408* [15]. Detection of these miRNA* sequences, either alone (miR167*, miR168*, and miR390*) or in addition to the corresponding miRNA guide strand (miR158 and miR408), shows highly selective PEV packaging of specific miRNAs. Such targeted uptake of specific miRNAs might be required to add an additional layer of regulatory complexity to ensure the correct abundance of each miRNA in *Arabidopsis* leaves. Furthermore, *Arabidopsis* leaf-generated PEV loading with specific miRNA* passenger strands may be to mediate their transport to distal tissues where they could convert into miRNA guide strands to direct miRISC-mediated regulation of target gene expression—target genes which are not transcriptionally active in the leaf tissues where the miRNA* strands are produced. As mentioned, Baldrich et al. [15] also identified an extracellular subpopulation of miRNAs which were not selectively loaded into the PEVs isolated from the intercellular wash fluid of *Arabidopsis* leaves, with this “PEV-free” miRNA subpopulation including miR165/166, miR167, miR398, miR844, miR848, and miR854. Considering that both the PEV and intercellular wash fluid samples were shown to be enriched for specific miRNAs, it is tempting to speculate whether the site of action of each of the detected extracellular miRNAs influences PEV loading? That is, the miRNA silencing signals which are required to direct their gene expression regulatory effects locally move from their site of synthesis to site of action in PEV-free form. In contrast, those miRNAs required to direct their gene expression regulation effects at tissues distal to their site of synthesis are transported to these systemic sites via a PEV-mediated mode of transport.

It is well known that plants use a range of signalling molecules such as hormones, metabolites, amino acids, and proteins as well as sRNAs to direct internal systemic communication. Furthermore, it is also well established that plants communicate externally with surrounding non-plant organisms such as rhizosphere bacteria and fungi via the secretion of volatile organic compounds including ethylene and jasmonates. In *Arabidopsis*, the influence of exogenous RNA on regulating gene expression in planta was assessed via hydroponically cultivating *MICRORNA* (*MIR*) gene overexpression lines next to wild-type *Arabidopsis* plants [123]. Specifically, *Arabidopsis* lines molecularly manipulated to overexpress the *MIR* genes *MIR156* and *MIR399* were cultivated in parallel with wild-type plants, and post-cultivation, the expression of the target genes of miR156 and miR399 was assessed in wild-type *Arabidopsis* plants to demonstrate the involvement of exogenously generated RNA in regulating *Arabdiospsis* growth and development. In several plant species, miR156 is thought to be a phloem-mobile miRNA [124]. In *Arabidopsis*, via controlling the expression of members of the *SQUAMOSA-PROMOTER BINDING PROTEIN-LIKE* (*SPL*) TF gene family, miR156 regulates several important developmental processes, including controlling the transition of *Arabidopsis* plants from juvenile to adult vegetative development [125]. Like miR156, the *Arabidopsis* miR399 is a phloem-mobile sRNA which is produced in aerial tissues and transported systemically to the roots where it controls the level of expression of its target gene *PHOSPHATE2* (*PHO2*) in low-phosphate (PO_4_) growth environments. Specifically, under low PO_4_ conditions, after its production in the shoots, miR399 is transported to the roots where it represses *PHO2* expression [126,127]. In turn, reduced PHO2 protein abundance, which functions as a ubiquitin-conjugating E2 enzyme, releases PHO2-directed repression of its regulated protein targets, several PO_4_ transporters. Therefore, in low PO_4_ conditions, the loss of PHO2-directed repression of the PO_4_ transporter, PHO1, leads to an increased uptake of inorganic phosphate from the soil and into the xylem of *Arabidopsis* roots [126,127]. In their study, Betti et al. [123] showed that in wild-type *Arabidopsis* plants cultivated in hydroponic media which was also used to cultivate the *Arabidopsis MIR156* and *MIR399* overexpression lines, miR156 and miR399 abundance was elevated, and the expression of the *SPL3* (miR156), *SPL9* (miR156), and *PHO2* (miR399) target genes was reduced. Via the use of a protoplast system, the authors went on to show that the higher degree of repression of miR156 and miR399 target gene expression observed in wild-type *Arabidopsis* plants was due to enhancement of the target transcript cleavage mode of RNA silencing catalysed by *Ath*-AGO1 [123]. The overproduction of a miRNA in one plant line and demonstration of its transport and functional activity in a second neighbouring plant line shows that (1) the transportation route must be highly targeted towards the neighbouring recipient organism, (2) the miRNA must be protected throughout its journey to maintain its functionality upon arrival at its destination, and (3) there must be a well-established uptake mechanism for the delivered miRNA to be taken up by recipient cells. PEVs, therefore, form the most suitable biomolecule able to perform all these distinct functions, functions which are essential for exogenous RNA delivery and functionality in plants. Unfortunately, the role of PEVs in directing the packaging, transportation, and delivery/uptake of *Ath*-miR156 and *Ath*-miR399 from the *Arabidopsis* transformant line in which they were overexpressed to a second recipient plant(s) was not assessed as part of the Betti et al. [123] study. Therefore, establishing the multifaceted roles of PEVs in this process represents an exciting avenue for future research.

In seed development, miRNAs have been shown to act as a crucial genetic regulator of both size and vigour and to regulate the chemical properties of seeds, including their nutritional composition [128,129]. In rice, the role of PEV-delivered miRNAs in regulating seed germination efficiency was assessed via comparison of the PEVs and their harboured miRNAs in (1) freshly harvested seeds, (2) freshly harvested seeds stored short term for 1 month, and (3) seeds which had been stored long term for a 3-year period under natural conditions [17]. This analysis showed that all three samples produced PEVs of a similar size range, ~70 to 210 nm, but that the predominant PEV size differed somewhat across the freshly harvested, freshly harvested and short-term-stored, and long-term-stored seed samples, with PEV size peaks of ~160, 138, and 143 nm reported, respectively [17]. The miRNA profile of the PEVs isolated from the three assessed samples differed much more considerably than did the diameter of isolated PEVs, with 187, 52, and 69 miRNAs catalogued for the PEVs extracted from freshly harvested, freshly harvested and short-term-stored, and long-term-stored seeds, respectively [17]. Of the miRNAs detected in all samples, ten miRNAs were determined to significantly differ in abundance between the three assessed samples, and these belonged to four highly conserved plant *MIR* gene families, including the *MIR159*, *MIR164*, *MIR166*, and *MIR168* gene families. After identification of miRNAs with altered abundance in the three assessed samples, the authors went on to next determine the functional roles of the putative target genes of *Osa*-miR159, *Osa*-miR164, *Osa*-miR166, and *Osa*-miR168 in rice. The target genes of *Osa*-miR159 were putatively determined to be involved in mitochondrial fusion and localisation regulation, general cell development, and the development of floral organ structures, including anther and pollen development [17]. The target genes of *Osa*-miR164 are likely involved in UDP-xylose metabolism and regulation of the activity of glycosyltransferases, and the target genes putatively under *Osa*-miR166-directed expression regulation were determined to be involved in lipid binding and various transportation functions, including regulation of transmembrane transport, plasma membrane export regulation, and regulation of the activity of transporters and antiporters. In addition to regulating the activity of *Osa*-AGO1, the key effector protein of the rice miRISC, *Osa*-miR168 was revealed to putatively regulate the expression of genes required for secondary shoot formation and the regulation of the peroxisome, a plant organelle essential for fatty acid reactions and the synthesis of the plant-growth-regulating phytohormones auxin and jasmonic acid [17]. Therefore, together, the functional annotation of the target genes putatively regulated by members of the four *MIR* gene families that differentially accumulated in PEVs isolated from rice seeds harvested and stored under three different treatment regimes showed that these PEVs were enriched in miRNAs that controlled molecular transport mechanisms and general plant development.

Like protein-coding genes, *MIR* genes are responsive to external environment stimuli, including alteration of *MIR* gene transcriptional activity in response to changes in light [130,131]. Light-induced alternation of *MIR* gene transcriptional activity is achieved via the interaction of light-responsive TFs binding to light-sensitive elements (*cis*-elements) in the promoter regions of *MIR* genes [130,131]. Specifically, via its interaction with photoactivated phytochrome B, the light-responsive TF, PHYTOCHROME INTERACTING FACTOR4 (PIF4), binds to its specific light-sensitive *cis*-elements in *MIR* gene promoters to alter the expressional activity of these *MIR* genes in response to changes to the wavelength or intensity of light in the growth environment [131,132,133]. Under high light, PIF4 alters the expression of *MIR165*/*166* loci, which in turn changes miR165/166 abundance. After its production, miR165/166 regulates the expression of specific members of the *HD-ZIP III* (class III homeodomain-leucine zipper) TF gene family, which play roles in plant development, such as controlling leaf polarity [134]. More specifically, miR165/166 accumulates to high levels on the abaxial (lower) epidermis of the leaf; however, its level gradually decreases towards the adaxial (upper) side of leaves. This gradient is formed via the intercellular movement of miR165/166, and the movement of this developmentally important miRNA is likely mediated via PEV transport [135]. Further to being involved in regulating leaf development, the miR165/166–*HD-ZIP III* expression module has more recently been shown to play a role in mediating phytohormone signalling, including auxin and abscisic acid signalling [136,137]. This finding shows that in addition to regulating plant development, miR165/166-directed gene expression regulation as plays a role in the response of a plant to abiotic stress, such as drought stress [136,137]. The role played by PEVs in mediating miR165/166 movement was assessed in maize (*Zea mays*) leaves sampled from plants cultivated in full light, red light, and the dark [16]. The authors grouped the isolated PEVs into two size classes, including PEVs with diameters larger than ~170 nm and PEVs of diameters less than 150 nm. In the larger-sized PEVs isolated from maize leaves sampled from plants cultivated under normal light, ~22% of *Zma*-miR165/166 was found inside vesicles (and ~78% was on the surface of these vesicles), whereas in this EV subclass isolated from maize leaves grown in the dark, only ~11% of the detectable *Zma*-miR165/166 pool was found inside vesicles. In direct contrast, the intravesicular content of *Zma*-miR165/166 almost doubled from ~30% in the smaller-sized PEV subclass isolated from maize leaves from light cultivated plants to over 55% for the smaller-sized PEV subclass isolated from maize plants cultivated in the dark [16]. This result not only showed that PEVs are likely involved in mediating *Zma*-miR165/166 movement in maize leaves but that further selectivity is placed on the PEV-mediated transportation of this miRNA via its selective packaging into a specific-sized subpopulation of maize PEVs, namely, those PEVs with diameters of less than 150 nm. Although current extraction techniques would not facilitate such location dependent analysis, it would be interesting to further this initial assessment to establish the internal-to-surface ratio of *Zma*-miR165/166 abundance for both PEV size classes in maize leaves isolated from the adaxial and abaxial tissues. Such an analysis could potentially allow for the full determination of the role played by PEV-mediated transport of *Zma*-miR165/166 from the underside (low light) to the upper surface (high light) of maize leaves—transportation which would help to establish the *Zma*-miR165/166 abundance gradient in this tissue for miR165/166 to direct its role in the normal development of maize leaves.

### 4.2. The MicroRNA Cargo of Plant Extracellular Vesicles and Their Role in Pathogen Defence

It is well established that both the host plant and invading pathogen secrete numerous molecules into extracellular spaces and beyond to direct defence and virulence processes, respectively, to facilitate bidirectional cross-border communication. Both plant- and pathogen-generated EVs would form effective conduits for the targeted delivery of these signalling molecules, with both fungal and bacterial invaders shown to promote the secretion of EVs by host plants, PEVs laden with defence-related lipids, metabolites, proteins, and sRNAs to be sent back to the invading pathogen “in reply” [138,139]. Such a central role for PEVs in pathogen defence is further supported by recent demonstrations showing that PEVs retain their bioactivity post-purification. Specifically, the growth of *Sclerotinia sclerotiorum*, including severe morphological changes and even cell death, was induced when this phytopathogenic fungus, which causes stem rot in agronomically important cropping species such as canola (*Brassica napus*), was treated with a PEV population purified from the extracellular fluid of sunflower (*Helianthus annuus*) seedlings [13]. Similarly, after their purification from the root system of tomato plants, PEV preparations were shown by De Palma et al. [112] to effectively retard the developmental progression (e.g., suppression of spore germination and inhibition of germination tube elongation) of several plant pathogens, including *Alternaria alternata* (black spot disease), *B. cinerea* (grey rot), and *Fusarium oxysporum* (fusarium wilt).

Facilitation of a route for the delivery of defence-mediating molecular signals by PEVs was once again supplied via use of the genetic model plant species *Arabidopsis*. As stated above, *Arabidopsis* cells secrete PEVs that are packaged with a population of preferentially loaded sRNAs at sites of *B. cinerea* infection [140]. Of particular interest is the demonstration that at infection sites, the *Arabidopsis*-generated PEVs are internalised by *B. cinerea*. Furthermore, after PEV internalisation and the release of PEV contents, the *Arabidopsis*-derived sRNAs suppress the expression of genes crucial for *B. cinerea* pathogenicity [140]. Also in *Arabidopsis*, PEVs have been shown to additionally transport the pathogen resistance proteins PENETRATION1/SYNTAXIN121 (PEN1/SYP121), PENETRATION3 (PEN3), and RESISTANCE TO POWDERY MILDEW8.2 (RPW8.2) to sites of infection by the powdery-mildew-causing fungus *Goloyinomyces orontii*. Moreover, all three *Arabidopsis* proteins subsequently accumulate in haustorial encasements to create a physical barrier for defence against *G. orontii* via impediment of its entry into surrounding *Arabidopsis* tissues [78,141]. In barley, infection by another powdery-mildew-causing fungus, *Blumeria graminis* f. sp. *hordei*, has been shown to induce MVBs to initially fuse to the plasma membrane and to subsequently release their internalised contents into the extracellular space in the form of paramural vesicles (EVs which specifically accumulate between the plasma membrane and cell wall of a plant cell) [142]. Once released, the paramural vesicles release their contents at *B. graminis* f. sp. *hordei* infection sites for the rapid deposition of cell wall appositions in the form of papilla structures [142]. Together, these works show that in addition to ferrying molecular-based defence signals to sites of pathogen infection, PEVs also harbour structural components (which are possibly pre-assembled) to be delivered to pathogen infection sites for the rapid construction of physical impediments to pathogen infection as a further defence mechanism.

Considerable evidence has now been accumulated to show that plant-derived miRNAs are transferred from the cells and tissues where their biogenesis occurs in the host plant to invading bacterial, fungal, and insect pathogens. One of the earliest reports of plant miRNA transfer to an invading pathogen came from the cotton/melon aphid *Aphis gossypii* [143]. Specifically, after feeding on melon (*Cucumis melo*), the aphid sRNA population was assessed via RNA sequencing. In addition to detecting known insect and animal miRNAs, together with the identification of aphid-specific miRNAs, numerous melon-derived miRNAs were detected in aphid samples, including members of the conserved plant *MIR* gene families *MIR156*/*157*, *MIR166*, *MIR168*, *MIR2911*, and *MIR2916* [143]. Interestingly, three of the plant miRNAs detected, namely, miR156/157, miR166, and miR168, previous to the Sattar et al. [143] study, had been shown to be present in the phloem sap of plant species such as apple (*Malus domestica*), canola, lupin (*Lupinus* spp.), and pumpkin (*Cucurbita maxima*). The detection of these three miRNAs in the aphid samples analysed by Sattar et al. [143] indicated that (1) these three miRNAs are also likely to represent phloem sap mobile miRNAs in melon, and (2) they are likely ingested by aphids during their feeding on melons.

Sorghum and barley are two monocot grass crops that are also vulnerable to aphid attack, with both widely cultivated species serving as a food source for the green bug (*Schizaphis graminum*) and yellow sugarcane (*Sipha flava*) aphid species [144]. Profiling of the sRNA landscape of these two aphid species not only identified known insect miRNAs and an additional subset of novel aphid-specific miRNAs but also detected eight known sorghum miRNAs, along with five and three novel miRNAs which mapped to the sorghum and barley genomes, respectively [144]. However, due to the application of an animal miRNA target gene identification strategy (i.e., only relying on “seed region” complementarity matches corresponding to miRNA nucleotide positions two to seven), thousands of putative aphid target genes were predicted for the 16 plant-derived miRNAs identified. Despite such an extensive list of putative aphid target genes being identified for the sorghum- and barley-derived miRNAs, GO term analysis did reveal enrichment for genes which encode proteins known to perform functional roles in cell signalling, cell-to-cell communication, and cell development [144]. This predictive analysis suggested that if functionally active in aphids, the plant-derived miRNAs would be able to disrupt signal transduction cascades and general development to influence overall aphid fitness.

Silkworms are the larvae of the silk moth *Bombyx mori*, with the larval stage of silk moth development being of particular economic value, as this is when silk is produced. As an oligophagous species, silkworms almost exclusively feed on the leaves of white mulberry trees or on the leaf material of other mulberry species. Mapping of the sRNA reads detected in the hemolymph of silkworms revealed the presence of eight mulberry-derived miRNAs, including *Mal*-miR156c, *Mal*-miR159a, *Mal*-miR166b, *Mal*-miR166c, *Mal*-miR167e, *Mal*-miR169a, *Mal*-miR396b, and *Mal*-miR398 [145]. Further analysis revealed that in addition to being present in the hemolymph, the *Mal*-miR166b, *Mal*-miR166c, *Mal*-miR167e, and *Mal*-miR396b silencing signals were detected in other silkworm body organs such as the fat body, silk gland, salivary gland, brain, gut, and sexual organs. This finding indicated that mulberry miRNAs are not just taken up by silkworms when feeding on mulberry leaves but that the ingested mulberry miRNAs are then delivered to, and taken up by, multiple silkworm tissues and/or cell types [145]. The authors of the study [145] did, however, go on to surprisingly show that although the mulberry-derived miRNA *Mal*-miR166b/c accumulated in multiple silkworm organs and the hemolymph, *Mal*-miR166b/c did not seem to execute any regulatory influence on the expression of any of the genes to which this mulberry miRNA displayed complementarity to.

The transfer of plant-derived miRNAs into another moth species, *Plutella xylostella* (diamondback moth or cabbage moth), has also been explored via sRNA sequencing. Worldwide, cabbage moth is a destructive pest of cruciferous vegetables such as cabbage, broccoli, brussels sprout (*Brassica oleracea* var. *gemmifera*), cauliflower (*Brassica oleracea* var. *botrytis*), and bok choy (*Brassica rapa* subsp. *chinensis*). Therefore, Zhang et al. [146] initially used sRNA sequencing to identify members of plant *MIR* gene families known to be highly abundant in the Brassicaceae. Post-application of stringent filtering parameters, the authors [146] confidently identified 21 known and 3 novel miRNAs in the hemolymph of *P. xylosetta* larvae which mapped to the *A. thaliana* genome (*Arabidopsis* is a member of the Brassicaceae). Furthermore, each of the 24 *Arabidopsis* miRNAs identified displayed the predominant characteristics of *Ath*-DCL1-generated miRNAs, being 21 nt in length and expressing a U residue as the 5′ terminal nucleotide. The presence of the three most highly abundant *Arabidopsis* miRNAs, specifically *Ath*-miR159a, *Ath*-miR166a, and *Ath*-miR7703, in *P. xylosetta* larval hemolymph was confirmed via a stem-loop PCR-based cloning approach. The use of in silico analyses next identified 12, 1, and 8 putative target genes in *P. xylosetta* for *Ath*-miR159a, *Ath*-miR166a, and *Ath*-miR7703, respectively, with three of the identified *P. xylosetta* putative targets belonging to the arthropod hemocyanin gene superfamily [146]. Via the use of an agomir approach (agomir: synthetic mimic of an endogenous miRNA), *Ath*-miR7705 was shown to (1) downregulate the expression of its targeted gene, *PROPHENOLOXIDASE2* (*PPO2*), and (2) negatively influence the development of *P. xylosetta*. More specifically, *Ath*-miR7703 agomir treatment shortened the larval stage while prolonging the pupal stage of *P. xylosetta* development, as well as to increase the incidence of development abnormalities expressed by treated pupae, and to also reduce the rate of adult moth emergence [146]. Together, the comprehensive findings presented in the Zhang et al. [146] study provided compelling evidence that host plant-derived miRNAs are indeed transferred to pathogens and that post-transfer, specific plant miRNAs maintain their functional activity to regulate pathogen gene expression, thereby reducing pathogen fitness.

The abundance of members of three highly conserved plant *MIR* gene families, including the miR156, miR159, and miR169 silencing signals, was determined for samples obtained from beehive structures as a first step towards showing that plant miRNAs highly abundant in the tissues of numerous plant species, and especially in pollen and nectar, could potentially be transferred to honey bees [147]. Interestingly, after the removal of the gut tissues from forager and nurse honey bees (*Apis mellifera* subspecies), only the miR156 sRNA could be detected in abdominal tissue in spite of the high levels of miR159 and miR169 detected for the beehive samples. This result showed that although numerous plant-derived miRNAs are transferred to honey bees from pollen and nectar during the feeding process and then subsequently transferred to various structures of the hive, only specific miRNAs are transported to other internal honey bee tissues following ingestion. It is important to note here that the authors go on to state that the level of detectable plant-derived miR156 is so low in internal honey bee tissues that its lowly abundance would likely prohibit the ability of miR156 to exert a biologically relevant regulatory effect on the expression of any honey bee transcript with an appropriate degree of complementarity to this plant-derived miRNA [147].

The demonstration of a lack of regulatory effect of dietary-consumed plant-derived miRNAs in honey bees was reinforced by a subsequent study [148] by the same team, a study which once again focused on the pollen-enriched miRNA miR156. As reported in the original study [147], the level of miR156 was compared between the midgut and abdominal tissues of forager and nurse bees [148]. This showed that compared with the modest levels of miR156 found in honey bee midguts, only very low levels of miR156 could be detected in abdominal tissues. In addition, via analysis of honey bees where the food bolus had been removed, an even further reduced level of miR156 was detected in the digestive system of these animals, which indicated that the higher level of plant-derived miR156 in the midgut resulted from ingested pollen itself and did not represent targeted delivery of this plant-derived miRNA to a specific tissue type or organ of honey bees [148]. Moreover, the abundance of miR156 in the systemic tissues of honey bees was more than 50-fold less than the moderately expressed endogenous miRNA miR277. This lowly abundance of miR156 led the authors to once again propose that miR156 is highly unlikely to be biologically relevant in any internal honey bee tissue unless aided by a highly specialised mechanism for concentrating or transferring this and/or other plant-derived miRNAs to specific cell types. However, the existence of such a mechanism to elevate plant-derived miRNAs to biologically relevant levels in specific internal honey bee tissues to biologically relevant levels seems unlikely considering the extensive set of experimental analyses reported in [147,148].

Cotton wilt disease is one of the most devastating diseases for many cotton-producing countries and is caused by a number of fungal species, including the asexual, soilborne hemibiotrophic species *Verticillium dahliae* Kleb. In host plant species, *V. dahliae* localises its colonisation in, or directly adjacent to, vascular tissues. Subsequently, once conidia and microsclerotium develop, the xylem of the host plant becomes blocked, leading to the expression of wilt disease symptoms. A sequencing approach was initially used to establish the sRNA landscape of hyphae sampled from *V. dahliae* following its infection of cotton (*Gossypium* spp.). This approach not only catalogued the *V. dahliae* sRNA population but also detected the presence of 28 previously identified cotton miRNAs [149]. Of the cotton miRNAs detected, miR159 and miR166 were among the most abundant with the authors going on to show that the abundance of both highly conserved plant miRNAs is also elevated in *V. dahliae* hyphae following its infection of *Arabidopsis* and tomato plants. This finding readily suggests that both plant miRNAs form targeted sRNAs which are specifically transferred from a range of host plant species to the invading fungus [149]. Accordingly, *V. dahliae* target genes with high degrees of complementarity to miR159 and miR166 were identified, including *Ca^2+^-dependent cysteine protease calpain 1* (*Vda-Clp-1*) and *Isotrichodermin C-15 hydroxylase* (*Vda-HiC-15*), respectively. Molecular analyses showed that miR159 and miR166 levels increased and the expression of *Vda-Clp-1* and *Vda-HiC-15* decreased in *V. dahliae* hyphae following its infection of both cotton and *Arabidopsis* plants. This finding demonstrated that these two plant-derived miRNAs are selectively exported to *V. dahliae* to direct RNA silencing of specific genes of this fungal pathogen [149]. The requirement of *Vda*-Clp-1 and *Vda*-HiC-15 for *V. dahliae* pathogenicity was elegantly demonstrated by Zhang et al. [149] via their demonstration that when cotton plants were inoculated with *V. dahliae* strains engineered to express miR159- and miR166-resistant versions of *Vda-Clp-1* and *Vda-HiC-15* (strains *Vda*-Clp-1m and *Vda*-HiC-15m), respectively, wilt symptoms appeared earlier. In addition, the severity of disease symptoms was also greater in this group of cotton plants than it was in cotton plants exposed to the unmodified strain of *V. dahliae*.

Significant economic losses to global grain production occur annually due to the cereal disease Fusarium head blight (FHB; Fusarium ear blight (FEB); scab), caused by the ascomycetous fungus *Fusarium graminearum* (*F. graminearum*). The wheat (*Triticum aestivum*) miRNA *Tae*-miR1023 formed one of the miRNAs which could be mapped back to the wheat genome following the analysis of sRNA libraries prepared from *F. graminearum* following its infection of wheat leaves [150]. In silico analyses revealed that the *F. graminearum* gene *FGSG_03101*, which encodes an a/b-Hydrolase protein thought to be important for *F. graminearum* pathogenicity, housed a highly complementary target sequence to *Tae*-miR1023. A modified virus-based approach was used to initially overexpress *Tae*-miR1023 on wheat leaves prior to inoculating these pre-treated leaves with *F. graminearum*. The lesion size and the number of *F. graminearum* spores were significantly reduced on *Tae*-miR1023-overexpressing leaves compared to leaves treated with an empty vector control prior to their inoculation with the same strain of *F. graminearum* [150]. Furthermore, molecular analyses revealed *Tae*-miR1023 abundance to be elevated and the expression of the a/b-*Hydrolase* gene to be reduced in *F. graminearum* used to inoculate *Tae*-miR1023-overexpressing leaves. Further evidence that *Tae*-miR1023-directed expression regulation controls the pathogenicity of *F. graminearum* was provided by [150] via their generation of a knockout mutant of the *F. graminearum* a/b-*Hydrolase* gene targeted by *Tae*-miR1023. Specifically, wheat leaves infected with this mutant strain of *F. graminearum* displayed reduced disease symptom severity, as shown for wheat leaves pre-treated with the *Tae*-miR1023-overexpression virus-based construct. Together, the findings reported in the study [150] show that upon infection by *F. graminearum*, wheat plants generate *Tae*-miR1023 and transfer this sRNA-silencing signal to *F. graminearum*, thereby reducing the pathogen’s fitness.

In wine grapes (*Vitis vinifera*), the necrotrophic fungus *B. cinerea* causes “botrytis bunch rot” and in horticultural crops, *B. cinerea* infection causes the common fungal disease “Grey Mold”, with *B. cinerea* related diseases predicted to cost up to $100 (USD) billion worth of produce loss annually [151]. In a groundbreaking study [12] that used *Arabidopsis* as the target species, 42 *Arabidopsis*-generated sRNAs were identified in *B. cinerea* after its infection of *Arabidopsis* leaves, including miR166 as well as members of closely related and plant-specific sRNA species, *trans*-acting small-interfering RNAs (tasiRNA). Selectivity of the *Arabidopsis*-generated sRNAs taken up by *B. cinerea* was provided by demonstration of the highly enriched abundance of siR483, and not of siR585, despite both tasiRNAs being derived from the same tasiRNA precursor transcript (i.e., the *Arabidopsis TAS1C* dsRNA). Similarly, only siR453 from the *Arabidopsis TAS2* dsRNA precursor was detected in *B. cinerea* samples post-infection, even though the *TAS2*-derived tasiRNA siR710 is over 30 times more abundant than siR453 in the *Arabidopsis* leaves onto which *B. cinerea* was inoculated [12]. These three *Arabidopsis*-derived sRNAs (miR166, siR453, and siR483) were further shown to be loaded into the *Arabidopsis* leaf-generated PEVs as they could only be detected after the PEVs were first lysed via detergent treatment prior to sRNA detection. These findings provided the first definitive example of plant-derived sRNAs being loaded into PEVs for their delivery to an invading pathogen. To confirm that this was indeed the case, Cai et al. [12] localised known MVB marker proteins to sites of *B. cinerea* infection on *Arabidopsis* leaves, sites to which the well-characterised exosome marker protein TET8 was also localised. Furthermore, an *Arabidopsis tet8 tet9* double mutant was generated via downregulating *TET9* expression in the *tet8* single mutant background using *TET9*-targeting artificial miRNA (amiRNA) technology, with the resulting *tet8 tet9* double mutant plants displaying a pronounced susceptibility to *B. cinerea* infection [12]. Finally, the authors demonstrated that the expression of the target genes of the *Arabidopsis* tasiRNAs siR453 and siR483 were downregulated in *B. cinerea* post its infection of *Arabidopsis* leaves, and that this downregulated target gene expression was due to the transcript cleavage mechanism of RNA silencing directed by the AGO proteins of the fungus [12].

In a second *B. cinerea*-focused study, two tomato-derived miRNAs, *Sly*-miR482 and *Sly*-miR1001, were initially shown to be decreased and increased in abundance, respectively, post-infection of tomato plants by the fungus [152]. To investigate the antifungal activity of *Sly*-miR1001, synthetic versions of miR1001 and of the miR1001/miR1001* duplex were added to *B. cinerea* conidiospore (reproductive structures of filamentous fungi) cultures prior to inoculation of tomato leaves. The addition of either the single- or double-stranded form of synthetic miR1001 to *B. cinerea* conidiospores was initially demonstrated to inhibit germination. Namely, germ tubes emerged from untreated *B. cinerea* conidiospores within 24 h of culturing, whereas miR1001 or miR1001/miR1001* treated conidiospores failed to produce germ tubes [152]. Similarly, in leaf inoculation assays, untreated *B. cinerea* conidiospores produced large necrotic regions, whereas the conidiospores treated with miR1001 and miR1001/miR1001* produced next to no necrosis, or they produced areas of necrosis with largely reduced diameters. Of significant interest from this work was the repeated demonstration that the duplexed form of miR1001 (miR1001/miR1001*) mediated a much greater inhibitory effect on *B. cinerea* virulence than did the mature version (miR1001) of this tomato miRNA. In addition, via degradome analysis, the authors identified two putative *B. cinerea* target genes for *Sly*-miR1001, including an *ATP-dependent metallopeptidase* (*Bcin03g02170.1*) and a *Cysteine-type endopeptidase* (*Bcin10g01400.1*). RT-qPCR-based expression analysis was next used to confirm that the tomato-generated miRNA, *Sly*-miR1001, downregulates the expression of both candidate genes in *B. cinerea*. Taken together, the elegant findings presented in the Meng et al. [152] study show that in response to *B. cinerea* infection, tomato generates and delivers *Sly*-miR1001 to the fungus to suppress the germination and infection ability of *B. cinerea* conidiospores by inhibiting the expression of *Bcin03g02170.1* and *Bcin10g01400.1* via a target transcript cleavage mechanism of miRNA-directed RNA silencing.

In *Arabidopsis*, a small number of miRNAs, like miR173, can trigger secondary siRNA production following cleavage of the target transcript, such as miR173-directed cleavage of the tasiRNA precursors, *Ath-TAS1* and *Ath-TAS2* [153]. Similarly, in *Arabidopsis*, miR161-directed cleavage of 13 of its *PENTATRICOPEPTIDE REPEAT* (*PPR*) target transcripts triggers the production of a population of secondary siRNAs. This process has been shown to be induced by the infection of *Arabidopsis* by *Phytophthora capsica* (*P. capsica*), a fungus-like pathogenic oomycete that causes blight and fruit rot of numerous important cropping species. Moreover, *P. capsica* infection of *Arabidopsis* was shown to induce *MIR161* expression to increase both the abundance of miR161 and that of two *PPR*-derived siRNAs, *PPR*-siRNA1 and *PPR*-siRNA2 [154]. *Phytophthora capsica*, like other filamentous pathogens, establishes a symbiotic relationship with its host plant via the formation of haustoria, invaginated structures that extend from hyphae and facilitate nutrient delivery from the host, as well as forming an interface for *P. capsica* to deliver effectors to the host [155]. To establish whether the miR161-triggered *PPR*-siRNAs could be transported to *P. capsica* haustoria, RT-qPCR was used to analyse the abundance of *PPR*-siRNA1 and *PPR*-siRNA2 in *Arabidopsis* PEVs. RT-qPCR showed that *PPR*-siRNA abundance was indeed elevated in *Arabidopsis* PEVs following *P. capsica* infection. Furthermore, the *P. capsica* gene *PHYCA_554980*, which encodes a constitutively expressed U2-associated splicing factor and which is targeted by up to six *Arabidopsis PPR*-siRNAs, was shown to have downregulated expression in *P. capsica*. Specifically, in *P. capsica* which had been used to infect an *Arabidopsis* transformant line molecularly modified to overexpress miR161, an increased abundance of both the triggering miRNA (miR161) and the resulting *PPR*-siRNAs was detected [154]. Further evidence that *Arabidopsis* sRNAs after their PEV-mediated delivery to *P. capsica* could control gene expression in the invading pathogen was also provided by the authors, who infected other *Arabidopsis* mutant lines defective in secondary siRNA production with *P. capsica*. Specifically, *PHYCA_554980* expression was revealed to be higher in *P. capsica* used to infect these *Arabidopsis* mutant lines than was the level of *PHYCA_554980* expression in *P. capsica* following its infection of wild-type *Arabidopsis* plants [154]. Like the results reported for the *Arabidopsis*–*B. cinerea* pathosystem [12], the findings reported in this elegant study [154] provide further substantive support for the notion that specific plant-produced sRNAs are preferentially loaded into PEVs to mediate their delivery to the invading pathogen and control pathogen gene expression as part of the plant’s defensive response.

The role of cross-kingdom miRNA-directed gene expression regulation has also been investigated for a second *Phytophthora* species, *P. infestans*, which causes “late blight” in potato, tomato, and other members of the Solanaceae. Moreover, *P. infestans* infection of potato was shown to alter the expression of a large number of known (*n* = 43) and novel (*n* = 128) potato miRNAs [156]. Known miRNAs, such as *Stu*-miR166, *Stu*-miR394, *Stu*-miR396, and *Stu*-miR6149, were bioinformatically predicted to putatively target *P. infestans* genes for expression regulation. Via the use of the *Nicotiana benthamiana* (*N. benthamiana*) leaf infiltration assay, the transient expression of the precursor transcripts of *Stu*-miR166, *Stu*-miR396, and *Stu*-miR6149 together with a *P. infestans* inoculum was shown to promote *P. infestans* colonisation of *N. benthamiana* leaves. In contrast, the expression of the *Stu*-miR394 precursor transcript *PRE-MIR394*, along with the *P. infestans* inoculum, was revealed to inhibit the ability of *P. infestans* to infect *N. benthamiana* leaves [156]. Using the same *N. benthamiana* leaf infiltration assay, Lou et al. [156] subsequently showed that the transient co-expression of *PRE-MIR394* with its *P. infestans* putative target gene, which encodes a Fibronectin type III protein, led to miR394-directed expression regulation of the *P. infestans* target gene [156]. The stable transformation of potato with a *Stu*-miR394 overexpression construct and the characterisation of the sRNA population of the PEVs generated by this transformant line upon *P. infestans* infection is, however, required to fully establish the degree of the role played by PEV-mediated delivery of miR394 to *P. infestans* to control the infection status of this oomycete pathogen.

### 4.3. Do Pathogen-Generated Extracellular Vesicles Deliver MicroRNA Effectors to Host Plants?

An expansive range of organisms have been shown to transfer their sRNAs to other species, including pathogen sRNA transfer to both plant and animal hosts [91,135]. However, the interorganism transfer of this crucial class of regulatory RNA has only been shown to be facilitated by EV-mediated trafficking in a small number of cross-kingdom relationships [12,14,91]. For plant pathogens, EVs could serve as a highly effective conduit for enhanced pathogenicity by targeting the delivery of nucleic acids (e.g., sRNAs), metabolites, and proteins to host cells, which, once delivered, would function as effectors to facilitate host colonisation. For example, the Gram-negative obligate aerobic bacterium *Xanthomonas campestris* which causes “black rot” disease in cruciferous vegetables has been shown to generate EVs that fuse directly with the plasma membrane of *Arabidopsis* cells during the initial stages of its infection [157]. Similarly, when cultured on plant tissues, the phytopathogens *B. cinerea*, *Blumeria hordei*, *Zymoseptoria tritici*, *F. graminearum*, *F. oxysporum*, and *Ustilago maydis* have all been shown to generate EVs which harbour a diverse molecular cargo consisting of species-specific formulations of metabolites, lipids, polysaccharides, proteins, and nucleic acids [158,159,160]. When considered together, these findings identify the pressing need to advance our current understanding of the degree to which EV-mediated delivery of phytopathogen-generated sRNAs is involved in the infection process.

One of the first reports on sRNA transfer from pathogen to host plant came from sRNA profiling of *Arabidopsis* and tomato plants post-infection with the grey mould disease causing fungus *B. cinerea* [151]. Profiling of sequencing data identified over 800 sRNAs that could be mapped back to the *B. cinerea* genome in both *Arabidopsis* and tomato samples. Using the stringent target gene identification criteria for plant miRNAs (i.e., near perfect miRNA–target gene complementarity), numerous *B. cinerea* sRNAs were matched to host genes present in both plant species, including members of the *MITOGEN-ACTIVATED PROTEIN KINASE* (*MAPK*) gene family [151]. In plants, MAPK-mediated signalling cascades modulate many aspects of plant development as well as to direct the response of a plant to either biotic or abiotic stress [161]. Therefore, due to their known role in plant immunity, the identified plant-encoded *MAPK* genes would form likely targets of *B. cinerea* sRNA pathogenicity effectors. Indeed, the putative *MAPK* target genes of some of the most abundant *B. cinerea*-derived sRNAs identified in both the *Arabidopsis* and tomato sequencing pools were reduced in their level of expression following *B. cinerea* infection [151]. The authors went on to generate stable transformant lines with each *Arabidopsis* line molecularly modified to express one of the three most abundant *B. cinerea* sRNAs via the use of amiRNA technology. In addition to reduced *MAPK* gene expression, each of the three *Arabidopsis* amiRNA transformant lines showed enhanced susceptibility to *B. cinerea* infection, as evidenced by the rapid onset of pronounced disease symptoms. This finding strongly suggests that the *B. cinerea* sRNA-targeted, host-encoded genes play a central role in immunity against the fungus. Considering that most of the highly abundant *B. cinerea* sRNAs displayed plant-miRNA-like properties, including a length of 21 to 22 nt and a 5′ terminal U residue, Weiberg et al. [151] next used immunoprecipitation assays to show the preferential loading of *B. cinerea*-derived sRNAs into *Ath*-AGO1, the primary effector protein of the *Arabidopsis* miRNA pathway. Finally, the authors revealed that with respect to its ability to infect *Arabidopsis* and tomato, a *B. cinerea dcl1 dcl2* double mutant had greatly reduced virulence. Taken together, the extensive set of results presented in the elegant study [151] showed that *B. cinerea* produces sRNAs with plant miRNA-like properties which are transferred to the host plant upon infection to regulate the expression of select host immunity genes: an altered host plant gene expression profile which would promote the pathogenicity of the colonising fungus.

The highly destructive wheat pathogen *Puccinia striiformis* f. sp. *tritici* (*P. striiformis*), which causes wheat stripe rust (wheat yellow rust) in this cereal species, has also been reported to produce a pathogenicity-promoting sRNA which the fungus transfers to host plants during the infection process [162]. Moreover, sRNA sequencing of *P. striiformis* germ tubes and subsequent RNA folding analyses were combined to identify a highly abundant *P. striiformis* miRNA-like (milRNA) sRNA termed *Pst*-milR1. In addition to identifying the mature milRNA, the authors identified the precursor transcript from which *Pst*-milR1 is released, a transcript which could only be mapped to the *P. striiformis* genome, and not to any known *T. aestivum* genome or transcriptome sequence [162]. Bioinformatics was next used to identify wheat genes that could serve as putative target genes of *Pst*-milR1-directed expression regulation, an approach that identified the *PATHOGENESIS-RELATED2* (*PR2*) gene, *Tae*-*SM638*, which encodes a β-1,3-glucanase, as the most likely target. The authors [162] used the *N. benthamiana* leaf infiltration assay to confirm that, indeed, *Pst*-milR1 could direct expression repression of *Tae*-*SM638* via a target transcript cleavage mode of RNA silencing in planta. To further confirm that *Pst*-milR1-directed silencing of *Tae*-*SM638* promotes the pathogenicity of *P. striiformis*, wheat plants were pre-treated with a modified virus expression vector which reduced the ability of *P. striiformis* to produce *Pst*-milR1 [162]. This approach showed that for pre-treated leaves, *Pst*-milR1 abundance was decreased, *Tae-SM638* expression was increased, and the ability of *P. striiformis* to establish infection sites was inhibited. These findings show that the *P. striiformis*-produced milRNA, *Pst*-milR1, functions as an important pathogenicity factor by reducing host plant resistance. Moreover, reduced host plant resistance to the fungus is achieved by the *P. striiformis*-derived milRNA silencing signal directing expression repression of the immunity-related gene *Tae-SM638* in recipient cells.

The identification of genes encoding central pieces of protein machinery required for sRNA production, such as *DCL*- and *RNA-DEPENDENT RNA POLYMERASE* (*RDR*)-encoding loci in the genome of the obligate pathogen *Hyaloperonospora arabidopsidis* (*H. arabidopsidis*), which causes downy mildew in *Arabidopsis*, led Dunkar et al. [163] to determine whether oomycete-derived sRNAs acted as virulence factors to promote host parasitism. Via the use of a tagged version of AGO1, the catalytic core of miRISC, immunoprecipitation of tagged *Ath*-AGO1 and sequence analysis of its loaded sRNAs identified numerous *H. arabidopsidis*-derived sRNAs 21 nt in length and which expressed a 5′ terminal U residue. Previous research showed that the expression of *Arabidopsis* genes *WITH NO LYSINE KINASE2* (*Ath-WNK2*) and *APOPLASTIC ENHANCED DISEASE SUSCEPTIBILITY1*-*DEPENDENT3* (*Ath-AED3*) is reduced upon *H. arabidopsidis* infection. Furthermore, both *Ath-WNK2* and *Ath-AED3* were subsequently determined to form putative target genes of two of the most highly abundant *H. arabidopsidis* sRNAs detected in *Ath*-AGO1 immunoprecipitated samples [163]. Indeed, RT-qPCR-based assessments revealed *H. arabidopsidis*-generated sRNA abundance to be elevated, and *Ath-WNK2* and *Ath-AED3* target gene expression to be reduced, in wild-type *Arabidopsis* plants following *H. arabidopsidis* infection. To confirm that reduced *Ath-WNK2* and *Ath-AED3* expression was the result of *H. arabidopsidis* sRNA-directed, *Ath*-AGO1-catalysed RNA silencing, Dunkar et al. [163] repeated their expression analyses on the *Arabidopsis ago1* mutant. As expected, *Ath-WNK2* and *Ath-AED3* expression remained at control levels (i.e., mock-infected plants) in *Arabidopsis ago1* mutant plants following *H. arabidopsidis* infection. Further confirmation that *H. arabidopsidis* sRNAs hijack AGO1 activity to facilitate its infection of *Arabidopsis* was provided by the authors’ demonstration that the *ago1* mutant was more resistant to *H. arabidopsidis* infection displaying a distinct disease phenotype compared with wild-type *Arabidopsis* plants. In addition, the disease phenotype displayed by the *ago1* mutant was also distinct from those expressed by the *Arabidopsis ago2* and *ago4* mutants following challenge with *H. arabidopsidis. Arabidopsis* AGO2 and AGO4 have both been shown to be involved in other sRNA-directed RNA silencing pathways in *Arabidopsis*. Therefore, together, the results of this *H. arabidopsidis* assay series showed that only *Ath*-AGO1 is involved in the *H. arabidopsidis* infection process and that other *Arabidopsis* AGO proteins do not play a role in *H. arabidopsidis* sRNA-directed expression regulation of *Arabidopsis* genes upon *H. arabidopsidis* infection.

In tomato and other related species, *F. oxysporum* causes vascular wilt disease, with symptoms becoming highly expressed when the fungal hyphae move from the root cortex to xylem vessels [164]. Ji et al. [165] assessed the ability of the milRNA *Fol*-milR1 to act as a pathogenicity factor of *F. oxysporum* f. sp. *lycopersici* (*Fol*) during its infection of tomato. Moreover, sRNA sequencing was initially used to identify *Fol*-derived milRNAs, and of the seven *Fol*-milRNAs identified, the 23 nt *Fol*-milR1 was determined to be the most abundant milRNA, accumulating to readily detectable levels in two tomato cultivars post-*Fol* infection. A gene replacement approach was next used to generate novel *Fol* strains which overexpressed and knocked out the encoding locus of *Fol*-milR1, with the growth and development of both *Fol* strains identical to that of the unmodified wild-type strain [165]. However, tomato infection assays demonstrated that modulation of *Fol*-milR1 abundance altered the pathogenicity of the fungus, with the overexpression strain having enhanced pathogenicity in tomato while the knockout strain showed an attenuated ability to establish infection of the susceptible tomato cultivar Moneymaker [165]. Sequence analysis of the tomato genome revealed tomato locus *Solyc06g007430*, which encodes the Calcineurin B-like (CBL) protein FUSARIUM RESISTANCE GENE4 (*Sly*-FRG4), to form the most likely target of *Fol*-milR1-directed expression regulation. Via the use of the *N. benthamiana* leaf infiltration assay, *Fol*-milR1 was indeed shown capable of directing cleavage-based expression regulation of the *Sly-FRG4* transcript. Furthermore, the Moneymaker line which had been molecularly modified to harbour a loss-of-function allele of *Sly-FRG4* was significantly more susceptible to *Fol* infection, displaying more severe disease symptoms than those displayed by *Fol*-infected Moneymaker plants which had not been modified [165]. Via the use of tagged versions of the *Sly*-AGO1 and *Sly*-AGO4a proteins, Ji et al. [165] next demonstrated that due to its non-canonical length with respect to plant miRNAs which are predominantly 21 nt, the 23 nt *Fol*-milR1 was preferentially bound by *Sly*-AGO4a and not by *Sly*-AGO1. When considered together, the results reported in the Ji et al. [165] study strongly imply that, indeed, *Fol*-milR1 acts as a fungal pathogenicity factor to promote the ability of *Fol* to infect susceptible tomato cultivars. More specifically, after its transfer from *Fol* to host cells, *Fol*-milR1 is preferentially bound by *Sly*-AGO4a to direct *Sly*-AGO4a-catalysed cleavage of the *Sly-FRG4* transcript as part of the infection process of *Fol* in susceptible tomato cultivars such as Moneymaker [165].

The soilborne pathogenic fungus *V. dahliae* can infect over 400 plant species to cause verticillium wilt in many of these species, a disease characterised by vegetative tissue discolouration and the curling of stems and leaves. Following on from their demonstration that cotton produces and transfers the miR159 and miR166 sRNA silencing signals to *V. dahliae* upon pathogen infection to control the expression of genes involved in microsclerotium formation and hyphal growth [166], the authors used the *Arabidopsis*–*V. dahliae* pathosystem to show that the invading fungus is also capable of producing and transferring sRNAs to the host plant. Moreover, immunoprecipitation of a tagged version of *Ath*-AGO1 and sequencing of *Ath*-AGO1-bound sRNAs revealed numerous sRNAs to have *V. dahliae* origins. Target gene identification in *Arabidopsis* for *V. dahliae*-derived sRNAs bound by *Ath*-AGO1, identified a highly complementary target site for the 24 nt *V. dahliae* sRNA, VdrsR-1, in the precursor transcript *PRE-MIR157D,* just downstream of the stem arm from which the mature miR157d silencing signal is liberated [166]. RT-qPCR was next used to show that *PRE-MIR157D* abundance was reduced in *V. dahliae*-infected *Arabidopsis* and that the levels of the mature miRNA processed from this precursor was elevated, a finding which suggested that VdrsR-1-directed processing of *PRE-MIR157D* by *Ath*-AGO1-catalysed cleavage leads to enhanced miR157d production. RT-qPCR further revealed that in response to elevated miR157d accumulation in *V. dahliae*-infected *Arabidopsis*, the expression of two of its endogenous targets, namely *SPL13A* and *SPL13B*, was reduced, presumably via enhanced miR157d-directed *Ath*-AGO1-catalysed expression repression [166]. SPL13A and SPL13B, along with other SPL TF family members, are positive regulators of juvenile-to-adult vegetative development, and vegetative-to-reproductive development in *Arabidopsis* [167,168]. Therefore, when considered together, the results presented in the Zhang et al. [166] study show that upon infection of *Arabidopsis*, *V. dahliae* secretes the VdrsR-1 silencing signal into host cells to enhance the accumulation of miR157d via promoting the efficiency of *PRE-MIR157D* processing. In turn, the increased level of *Ath*-miR157d leads to decreased *Ath*-*SPL* target gene expression, an expression change which would delay the transition of infected *Arabidopsis* plants from vegetative to reproductive development to promote further propagation of host tissues by *V. dahliae*.

The ectomycorrhizae form a class of mutualistic fungi which establish essential nutrient-acquiring symbioses with the feeder roots of many temperate and boreal forest trees [169]. As part of the early stages of symbiosis, ectomycorrhizal (ECM) fungi transfer effector molecules to host cells to alter plant signalling and metabolism [170]. To date, only a small number of protein-based effectors have been characterised for ECM fungi [170]. More recently however, sRNAs have also been proposed to likely act as effectors for symbiosis establishment, with ECM fungi shown to possess all the necessary components for sRNA biogenesis, including encoding *AGO*, *DCL*, and *RDR* genes [171]. To establish a role for miRNA-directed gene regulation in host cells during ECM symbiosis, Wong-Bajracharya et al. [172] studied the mutualistic ECM fungus *Pisolithus microcarpus* (*P. microcarpus*) and its interaction with the roots of one of its host species, the hardwood eucalypt species *Eucalyptus grandis* (*E. grandis*). Sequencing of the sRNA population of *P. microcarpus* during colonisation of *E. grandis* roots allowed for the confident identification of eleven *P. microcarpus* milRNAs. The abundance of one of the identified milRNAs, *Pmic*-milR8, was induced during the in-growth of the mycorrhizal fungus into the apoplastic space of *E. grandis* roots [172]. Fluorescent in situ hybridisation confirmed that following its production by *P. microcarpus*, *Pmic*-milR8 accumulates in punctate structures within *E. grandis* root cells. The role of *Pmic*-milR8 in colonisation of *E. grandis* roots by *P. microcarpus* was further characterised via the use of a synthetic version of *Pmic*-milR8. This approach showed that this fungus-produced sRNA potentially acts as a positive regulator of Hartig net formation. In symbiosis fungi, the Hartig net is the network of inward-growing hyphae which extend from the hyphal mantle of the fungus and which penetrate between the cells of the root cortex of the host plant. Moreover, Wong-Bajracharya et al. [172] revealed a positive correlation between the level of supplementation with synthetic *Pmic*-milR8, and the degree of depth of penetration of the Hartig net in treated *E. grandis* roots. This finding led the authors to predict putative *E. grandis* target genes for *Pmic*-milR8, with this analysis identifying seven *E. grandis* targets, three of which were determined to form *Pmic*-milR8 targets of high confidence. Interestingly, this putative target gene identification analysis failed to identify any putative targets for *Pmic*-milR8-directed expression regulation in *P. microcarpus* itself, with only host genes identified as putative targets. RT-qPCR was next used to assess *E. grandis* target gene expression in *P. microcarpus*-treated *E. grandis* roots and revealed an expected expression trend of decreased target gene expression in *E. grandis* samples where the abundance of *Pmic*-milR8 was elevated [172]. Of the *E. grandis* target genes analysed by RT-qPCR, gene *Eucgr.E03170*, which encodes a CC nucleotide binding and leucine-rich repeat domain (CC-NLR) immune-related receptor protein, was revealed to be the most responsive to altered *Pmic*-milR8 abundance. Therefore, the Wong-Bajracharya et al. [172] study revealed that the ECM fungal species *P. microcarpus* produces specific milRNAs which control target gene expression in host cells, a form of milRNA-directed cross-kingdom gene expression regulation which likely aids in the establishment of the symbiotic relationship between the fungus and its plant host.

*Blumeria hordei* is an ascomycete fungus that causes powdery mildew in barley, with the biotrophic nature of this parasitic fungus requiring tight control of host gene expression to ensure progression of its own life cycle. Well over 1200 sRNA-producing loci have been identified in *B. hordei*, with just under half of these further shown to potentially have putative target genes in barley [173]. Of particular interest was the earlier observation of the formation of EV-like structures at the interface of barley leaf epidermis cells and *B. hordei* haustoria [10]. Together, these previous reports led Kusch et al. [174] to determine the degree of involvement of EV-mediated trafficking of sRNAs from *B. hordei* haustoria to barley leaves to control host gene expression. Interestingly, sRNA profiling of infected barley epidermis, *B. hordei* haustoria, and the EVs which accumulated at infection sites, revealed enrichment of specific sRNA classes derived from both host plants and infecting fungus. Specifically, sRNAs belonging to the rRF (ribosomal RNA (rRNA)-derived RNA fragments) and tRF (transfer RNA (tRNA)-derived RNA fragments) subclasses were the most enriched sRNA species derived from barley (epidermis and EV samples) and *B. hordei* (haustoria and EV samples), respectively [174]. Although the rRF and tRF subclasses of sRNA were shown to predominate the sRNA profile of each analysed sample, numerous milRNAs were identified amongst the specific *B. hordei* samples. In addition, target gene identification analysis further revealed that these *B. hordei*-derived milRNAs had a greater propensity to target barley genes for expression regulation than to target genes endogenous to the fungus. This finding indicates that the identified *B. hordei*-derived milRNAs are produced by the fungus to direct cross-kingdom RNA silencing in host cells at infection sites. GO term analysis of barley genes putatively targeted by *B. hordei*-derived milRNAs added further support to this suggestion with GO term enrichment revealed for functional categories such as “microtubule-severing ATPase activity” and “vacuole” for barley target genes [174]. Overall, the findings presented by Kusch et al. [174] show that of the large population of sRNAs produced by *B. hordei* upon barley infection, the milRNA class form the most likely candidates for sRNAs transferred to host cells via an EV-mediated transport route to regulate host gene expression promote *B. hordei* pathogenicity.

The first evidence of miRNA transfer as part of plant–plant interactions was provided by Shahid et al. [175] via their demonstration that the obligate parasitic plant *Cuscuta campestris* (*C. campestris*) transfers numerous miRNAs to the cells of the host plant which surround its haustoria. Moreover, 76 *C. campestris* sRNAs, including 43 miRNAs, were revealed to be more abundant in local *Arabidopsis* cells than in the haustoria which these cells surround, a finding which indicated selective transfer of specific *C. campestris* sRNAs into host cells [175]. Interestingly, 26 of the 43 *C. campestris*-derived miRNAs were 22 nt in length, a sRNA size in plants which triggers secondary siRNA production following initial transcript cleavage directed by the targeting miRNA [176]. Six *Arabidopsis* transcripts, including the partially redundant auxin receptors *TRANSPORT INHIBITOR RESPONSE1* (*TIR1*), *AUXIN-SIGNALLING F-BOX2* (*AFB2*), and *AFB3*, were identified as putative target genes of the 22 nt *C. campestris* miRNAs, and to which secondary siRNAs could be mapped via analysis of sRNA sequencing data [175]. It is important to note here that the endogenous miRNA, *Ath*-miR393, also targets *TIR1*, *AFB2*, and *AFB3* for cleavage-based RNA silencing and subsequent secondary siRNA production. However, the in-depth analysis of the sRNA sequencing data stemming from *C. campestris* parasitised *Arabidopsis* cells definitively showed that the secondary siRNAs detected were generated after *C. campestris* miRNA-directed cleavage of *TIR1*, *AFB2*, and *AFB3,* and were not produced following the cleavage of these three auxin-receptor-encoding transcripts by their endogenous targeting miRNA, *Ath*-miR393 [175]. Via repeating this analysis on a series of *Arabidopsis* mutant plant lines defective in the function of the machinery proteins required for secondary siRNA production (e.g., *Arabidopsis* knockout mutants *dcl4* and *rdr6*), the authors convincingly showed that *C. campestris*-derived miRNAs are active inside host cells, and further, hijack the endogenous sRNA silencing-machinery proteins to control the expression of host genes which are normally involved in mounting a defensive response against invasion by this parasitic plant species [175].

Whitefly (*Bemisia tabaci*) is a highly polyphagous plant pest, with more than 600 different plant species documented to be used by whitefly as a host [177]. Unsurprisingly, numerous agronomically important species are included in the expansive list of whitefly host species, with this insect pest now reported to have invaded and caused significant loss to agricultural production in over 100 countries worldwide [178]. Whitefly is a phloem-feeding herbivore which uses its needle-shaped mandibles to traverse the host plant epidermis to feed on the nutrient-rich phloem, a process which relies heavily on the secretion of saliva-delivered effectors [178]. To determine if EV-trafficked miRNAs act as whitefly effectors to facilitate feeding on host plants, Han et al. [14] assessed the abundance of whitefly miRNAs in the insect itself and on tobacco (*Nicotiana tabacum*) plants. The RT-qPCR-based assessment showed that although numerous whitefly miRNAs were readily detectable in whitefly tissues, only *Bta*-miR29b could be detected in the leaves and phloem sap of whitefly-infested tobacco. Confirmation that *Bta*-miR29b indeed acts as a whitefly-saliva-derived miRNA effector was supplied by the authors’ demonstration that *Bta*-miR29b was also detected in the leaves and phloem sap of *Arabidopsis* and cotton plants following whitefly infestation [14]. To further confirm that *Bta*-miR29b functions a saliva-delivered effector, the transient expression of a *Bta*-miR29b artificial target mimic (a non-cleavable decoy transcript which sequesters the miRNA) on tobacco leaves prior to whitefly infestation was shown to greatly reduce the fecundity and survival rate of whitefly. Furthermore, knocking down or knocking out the function of genes essential for EV formation, EV cargo sorting, and EV transport and secretion in whitefly prior to performing feeding experiments on tobacco additionally revealed an essential role for the EV system for the packaging and delivery of *Bta*-miR29b (and likely for other whitefly sRNAs also) from the pest to host plants [14]. After revealing that *Bta*-miR29b is delivered to tobacco cells via an EV-mediated mode of transport, Han et al. [14] performed sequence analysis to identify the most likely tobacco target of *Bta*-miR29b-directed expression regulation. The analysis identified the tobacco gene *Nta-BAG4* (*BCL-2-ASSOCIATED ATHANOGENE4*) as the most likely target of *Bta*-miR29b expression regulation. The *BAG* gene family is tightly conserved across plant species with family members shown to play key roles in growth and development, including senescence, and in mediating defence responses against abiotic and biotic stress [179]. Degradome sequencing showed that the *Nta-BAG4* transcript is cleaved within the *Bta*-miR29b target site in whitefly-infested tobacco. Furthermore, a role for *Nta-*BAG4 in promoting the defence of tobacco against whitefly infestation was established via the authors’ demonstration that whitefly fecundity and survival rates were reduced on tobacco plants engineered to transiently overexpress *Nta-BAG4* [14]. Together, the comprehensive series of elegant findings presented in the Han et al. [14] study show that the whitefly miRNA, *Bta*-miR29b, is packaged into the salivary EVs of the insect to mediate its delivery to plant cells upon feeding to silence the expression of *Nta-BAG4*, a host plant gene involved in mediating a defence response.

### 4.4. Algae Extracellular Vesicles: Do They Mediate a Similar Transportation Role in MicroRNA-Directed Gene Expression Regulation?

Like plants and animals, both micro- and macroalgae produce EVs, with algae viewed as a promising future EV source offering a rapid, natural, and sustainable route for EV production [180]. The bioactive cargo harboured by numerous marine and freshwater microalgal species has been characterised, with these works collectively showing that algal EVs (AEVs) likely mediate roles in growth and development, cellular communication, gene transfer, phage resistance, toxicity, and adaptation to environmental stress [181,182]. Using techniques such as TEM, Dynamic Light Scattering (DLS), and Nanoparticle Tracking Analysis (NTA), the AEVs derived from microalgae were determined to have an average diameter of 70–135 nm [183]. More specifically, *Synechocystis* sp., *Chlamydomonas reinhardtii*, *Alexandrium minutum*, and *Haematococcus pluvialis* have been shown to produce AEV populations of differing sizes, including vesicle diameter ranges of 24–450 nm, 37–710 nm, 10–660 nm, and 90–120 nm reported, respectively [181,182,183,184]. Interestingly, for *H. pluvialis*, AEV diameter appeared linked to development stage, with average AEV diameters of ~89, 112, and 118 nm determined for the green motile, green nonmotile, and red nonmotile cyst stages of its life cycle, respectively [182]. This finding is of particular interest as it not only infers species-specific variation in AEV size, but it also indicates that for some microalgae, AEV physical characteristics are dependent on developmental stage. For the microalgae *Tetraselmis chui* and *H. pluvialis*, AEV morphology ranges from globular, to quasi-spherical, through to completely spherical particles [182,183].

Macroalgae EVs appear to be slightly larger than those produced by their microalgae counterparts, with average diameters of ~150 and 200 nm reported for the AEVs produced by the green seaweed (Chlorophyta) *Codium fragile* and the brown seaweed (Phaeophyceae) *Sargassum fusiforme*, respectively [185]. TEM revealed that seaweed AEVs primarily adopt a spherical shape, and furthermore, this analysis confirmed that both the AEVs formed by micro- and macroalgae are enclosed by a membranous bilayer [182,183,185]. In addition, Adamo et al. [183] characterised the bilayer membrane of AEVs to uncover an enriched abundance of members of protein classes such as Alix (endosome–lysosome system regulator proteins), Enolase (membrane embedded plasminogen receptor proteins), HSP70 (molecular chaperone and protein folding proteins), β-actin (cell motility, cell structure, and cell integrity proteins), and H^+^-ATPase (active transmembrane transporter proteins). Together, the specific members of the protein classes identified [183] not only form suitable biomarkers for future AEV characterisation studies but also contribute to the zeta potential of −13 to −25 mV for AEVs—a zeta potential that would confer AEV stability in suspension.

Characterisation of the molecular cargo harboured by AEVs revealed similarities to the internal composition of PEVs, with AEVs also demonstrated to harbour rich payloads composed of lipids, proteins, and nucleic acids, including mRNAs, miRNAs, and other sRNA classes [186]. For example, the AEVs of the marine Chlorophyte microalgal species species *Tetraselmis chui* [183], harbour bioactive pigments including the high-value carotenoids, neoxanthin, violaxanthin, lutein, and β-carotene (all of which possess extensive antioxidant properties driving their expanding use in food, cosmetic, and pharmaceutical applications), as well as long-chain polyunsaturated fatty acids (PUFA) such as eicosapentaenoic acid (EPA; shown to prevent blood clotting, to reduce blood triglyceride levels, and which may also have anti-inflammatory and analgesic effects in humans). Proteomic analysis of *C. reinhardtii*-derived microalgal AEVs identified over 10,800 vesicle-harboured proteins, with 442 and 480 proteins shown to have significantly altered abundance via comparison of the protein profiles of AEVs isolated from control *C. reinhardtii* samples (isolated from *C. reinhardtii* during its plateau phase of growth) to those EVs isolated from *C. reinhardtii* samples following cultivation in nutrient-depleted conditions or the log phase of *C. reinhardtii* growth, respectively [184]. This comparative proteomic study facilitated the identification of key sets of *C. reinhardtii* proteins which are loaded into AEVs and known to mediate functional roles in stress responses and standard growth and development, either in *C. reinhardtii* itself or in other molecularly characterised microalgae species. Of note, amongst the differentially abundant proteins identified in *C. reinhardtii* AEVs were numerous flagella-associated membrane proteins, cell-wall-associated proteins, extracellular matrix proteins, and three *C. reinhardtii* ESCRT homologues, specifically proteins A8IXRO, A8IAJ1, and A0A2K3E4X9 [184]. Furthermore, via the use of fluorescence emitting lipophilic dyes, the authors of the extensive study [184] went on to show that labelled AEVs were readily transferred between, and internalised by individual cells, a finding which implies that like the in plant and animal systems, the EVs produced by microalgae form a transportation route for the external transfer of molecular information and/or molecular resources.

The AEVs produced by the small number of microalgae species characterised to date have been reported to harbour miRNAs as part of their extensive molecular cargo. In *C. reinhardtii*, high-throughput sequencing together with bioinformatics was used to identify 64 known miRNAs and to predict 84 novel miRNAs specific to *C. reinhardtii* [184]. Intriguingly, less than 7.5% of the miRNAs detected in the analysed AEVs could also be detected in *C. reinhardtii* host cells, a finding that strongly infers highly selective packaging of miRNAs into the AEVs produced by *C. reinhardtii* [184]. In both micro- and macroalgae, studies have repeatedly shown that miRNAs are not phylogenetically conserved, and furthermore, instead of primarily functioning as regulators of developmental gene expression, algae miRNAs appear more likely to play a role in regulating molecular-based responses to various forms of stress, such as adapting to nutrient deprivation [187,188]. Further support of this alternative primary regulatory role for miRNA-directed gene expression regulation in algae was offered by Li et al. [184], who showed that in response to growth in a sulphur-depleted environment, 16 of the 64 (25%) known miRNAs loaded by AEVs have previously been shown to mediate roles in the response of *C. reinhardtii* to an altered abundance of sulphur in the growth environment.

Profiling of the AEVs isolated from the three developmentally distinct stages of the life cycle of the microalga *H. pluvialis* (i.e., the green vegetative motile, green vegetative nonmotile, and red nonmotile cyst stage), identified 93 known and 70 novel miRNAs [182]. Many of the known miRNAs identified in *H. pluvialis* AEVs corresponded to known plant miRNAs, with these miRNAs shown to have an altered abundance in the AEVs isolated from the three different stages of *H. pluvialis* development analysed [182]. Bioinformatics was used to identify almost 1500 putative target genes for the 163 mature miRNAs identified in *H. pluvialis* AEVs, with ~140 putative targets identified for the ten most highly abundant miRNAs in *H. pluvialis* AEVs. Interestingly, functional categorisation of the putative target genes encoded by the host genome of the ten most abundant AEV miRNAs revealed enrichment in a range of functional categories, including chaperones, amino acid transport, lipid transport, protein transport, protein secretion, intracellular trafficking, vesicular transport, and the biogenesis of components of the cytoskeleton as well as the cell wall, cell membrane, and cell envelope [182]. This result suggests that many of the miRNA target genes grouped under these functional categories are involved in the EV production process in *H. pluvialis*. This also raises the intriguing possibility that in this freshwater green algal species, miRNAs may regulate the production of the vesicles into which they are subsequently loaded into to further regulate other aspects of *H. pluvialis* development, or to mediate its adaptive responses to an altered growth environment [182].

In the euryhaline microalga *Nannochloropsis oculata* (*N. oculata*), 182 known and 131 novel miRNAs were identified by Zheng et al. [189], and when the authors fed Akoya pearl oyster (*Pinctada fucata*) a pure *N. oculata* diet, 23 microalgae-derived miRNAs were isolated from Akoya pearl oyster haemolymph exosomes. This finding showed that algae-derived miRNAs can be transferred between species, and after their transfer from algae to oyster, most likely via oral ingestion, a select cohort of *N. oculata* miRNAs enter the circulatory system of this bivalve mollusc species via packaging into oyster haemolymph exosomes. Interestingly, distinct sets of putative target genes were bioinformatically identified in *N. oculata* and Akoya pearl oyster for the *N. oculata* miRNAs loaded into Akoya pearl oyster haemolymph exosomes [189]. Moreover, in *N. oculata* cells, target gene functional categorisation for the 228 putative candidates identified revealed functional enrichment for genes involved in cellular processes related to energy production such as ATPase activity, phosphorylation activity, hydrolase activity, and nucleoside-triphosphatase activity. In contrast, KEGG enrichment analysis of the almost 700 Akoya pearl oyster genes identified as potential targets of the *N. oculata* miRNAs harboured by haemolymph exosomes were enriched for genes involved in endocytosis-linked processes and various signalling pathways, such as the FOXO, MAPK, and AMPK signalling pathways [189]. Taken together, the studies conducted to date in *C. reinhardtii*, *H. pluvialis*, and *N. oculata* strongly indicate that like in the plant and animal systems, AEVs most likely form a conduit for the cross-kingdom transfer of miRNAs for the safe passage and delivery these regulatory molecules to targeted cells and/or tissues to mediate crucial roles in communication and signalling.

## 5. Future Directions: The Use of Extracellular Vesicles and Their miRNA Cargo in the Agricultural and Aquacultural Settings

The repeated demonstration of the rich repertoire of RNA molecules, including the miRNA class of small regulatory RNA, harboured by EVs isolated from numerous plant and algal species intimates a potential future role for use of these vesicles as natural delivery and storage vessels in agriculture and aquaculture. In recent decades, RNA interference (RNAi)-based technologies have not only proven to provide an exceptionally powerful tool to elucidate gene function in plants, but are demonstrated to also offer an efficient strategy to control numerous agronomically significant pathogens. However, in the environment, RNA molecules are highly unstable, being susceptible to rainfall, UV light, and high temperatures [190,191,192]. Furthermore, plants and their interacting organisms (regardless of the beneficial through to the pathogenic nature of the interaction) naturally secrete an array of enzymes extracellularly, which can also negatively impact RNA stability, stability which is further impacted by the differing efficiency of uptake of externally delivered RNA by each organism [60,190,193]. A new and innovative RNAi technology shown to form an effective plant disease management strategy, is SIGS: Spray-Induced Gene Silencing [12,60,190,194]. Although, as for naturally occurring extracellular RNAs, the efficiency of SIGS is highly influenced by environmental factors together with the degree of cellular uptake by the plant or pathogen species to which the RNA is topically applied. With respect to plant microbes, *Aspergillus niger*, *B. cinerea*, *Rhizoctonia sotani*, and *V. dahliae* readily take up RNA from the external environment, whereas *Colletotrichum gloeosporiodes*, *P. infestans*, and *Trichoderma virens* show modest to next to no uptake of externally delivered RNA [190,195]. To address these differences, their now repeated demonstration as vessels for RNA transport, identifies PEVs and AEVs as potential efficacious additives for SIGS formulations to (1) protect the RNA cargo, and (2) facilitate its efficient cellular uptake after its topical delivery. For example, Qiao et al. [190] showed that spraying naked dsRNA onto tomato protected the plants from *B. cinerea* infection for 7 days, whereas the same group [196] reported that grape was resistant to *B. cinerea* infection for 21 days when the dsRNA was encapsulated in artificial EVs.

In plants, long molecules of dsRNA form potent triggers of siRNA- and miRNA-directed RNA silencing; however, to date, RNAi technologies in plants have more frequently relied on the use of siRNA producing dsRNA triggers. Likely due to their perceived more complex design requirements and concerns over precursor-processing efficiency, amiRNAs have been less frequently used as an RNAi technology in plants. However, amiRNA technology does offer several distinct advantages over siRNA-based approaches, namely: (1) efficient precursor transcript processing; (2) high target specificity, and; (3) low “off-target” effects. Considering that miRNAs have now been catalogued in a vast array of genetically distinct plant species, and even more recently in a number of economically important plant pathogens, the precursor transcript, specifically the smaller-sized pre-miRNA processing intermediate, of a highly abundant miRNA in the target plant and/or pathogen (if known) species should be selected for molecular modification to generate the synthetic transcript for delivery of the amiRNA silencing signal. Such an approach largely rules out any concerns regarding low processing efficiency of the amiRNA from its precursor transcript, as the modified transcript will mimic the folding structure of the endogenous precursor, thus ensuring efficient recognition and processing of the precursor by the machinery proteins of the production stage of the miRNA pathway (e.g., SE1, DRB1, and DCL1). Such an approach will additionally ensure direct loading of the liberated amiRNA/amiRNA* duplex into miRISC for amiRNA selection by AGO1, the primary piece of protein machinery of the action stage of the plant miRNA pathway [119,120]. Similarly, if the amiRNA approach is to be used as the trigger of RNA silencing to offer the plant protection against an invading pathogen, then a miRNA precursor transcript endogenous to the plant pathogen itself can be used for amiRNA delivery. Such an approach would likely leave the amiRNA precursor in its folded and unprocessed state on the surface of the plant (or even if it were to be internalised by plant cells) and to only form a substrate for processing to generate the mature amiRNA post uptake and processing by the pathogen’s own miRNA pathway machinery proteins. This scenario would also direct considerably higher degrees of RNA silencing in the pathogen, as the amiRNA would not have been diluted by the plant’s own sRNA pool prior to pathogen uptake.

A second considerable advantage of amiRNA technology over dsRNA triggers, from which a large pool of sequence distinct siRNAs are generated, is target specificity. Moreover, only a single amiRNA silencing signal, usually 21 nt in length, is processed from the molecularly modified amiRNA precursor transcript. Most plant genes belong to multimember families, and therefore, the amiRNA approach allows for the design of a silencing signal which will only direct RNA silencing against a single, specifically targeted family member. Alternatively, the expression of a subclade, or even an entire gene family, can also be achieved by designing an amiRNA silencing signal that targets a highly conserved sequence, such as one that encodes a functional domain essential to the function of each family member. Such specificity in amiRNA design also greatly limits the concerns surrounding “off-target” effects, that is; altering the expression status of a gene(s) not intended to be a target of RNA silencing. Such off-target effects are hard to control when using a dsRNA trigger which is processed into a large and sequence diverse pool of siRNAs. Furthermore, siRNA-directed cleavage of some targets can identify the cleaved transcript as a template for dsRNA synthesis and secondary siRNA production, which further diversifies the sequence composition of the siRNA pool. In the model plant species *Arabidopsis*, most miRNAs that trigger secondary siRNA production are 22 nt in length rather than the canonical 21 nt length of most other *Arabidopsis* miRNAs. Therefore, an amiRNA precursor transcript from which only a 21 nt silencing signal is processed should be used to further limit the likelihood that amiRNA-directed cleavage of the targeted transcript results in secondary siRNA production.

One of the first demonstrations of the use of amiRNA technology to provide protection against plant pathogens was made in *Arabidopsis* [197]. Specifically, the precursor transcript of the highly abundant *Arabidopsis* miRNA miR159 was artificially modified to direct silencing against the silencing suppressor proteins (SSPs) p69 and HC-Pro of *Turnip yellow mosaic virus* (TYMV) and *Turnip mosaic virus* (TuMV), respectively. *Arabidopsis* transformant lines molecularly modified to overexpress amiR-p69 and amiR-HC-Pro were resistant to TYMV and TuMV infection, respectively [197]. Similarly, the genetic modification of tobacco with amiRNA generating transgenes which targeted the SSPs, p25 of *Potato virus X* (PVX) and HC-Pro of *Potato virus Y* (PVY), were resistant to infection by *PVX* and *PVY*, respectively [198]. Further, after the molecular modification of tomato plants to express amiRNAs which targeted specific sequences of the *Cucumber mosaic virus* (CMV) genome, including the 2a and 2b genes and 3′ UTR, amiR-2a-, amiR-2b-, and amiR-3UTR-expressing transformant lines were all shown to be resistant to CMV infection and to infection by the closely related viruses *Tobacco mosaic virus* (TMV) and *Tomato yellow leaf curl virus* (TYLCV) [199]. In rice, amiRNA technology combined with the use of tissue-specific promoters knocked down the expression of the dominant allele of *Xa13*, an elegant strategy which rendered the modified rice plants resistant to *Xanthomonas oryzae pathovar oryzae*, the causative agent of the highly devastating disease rice bacterial blight, while maintaining the full fertility of non-modified wild-type rice plants [200].

The amiRNA approach has also been used in rice to provide resistance to infection by the larvae of the moth *Chilo suppressalis* (striped rice stemborer). Moreover, rice transformant lines which were molecularly modified to overexpress the precursor transcript of the *C. suppressalis* miRNA, *Csu*-Novel-260, displayed considerable resistance to striped rice stemborer infection [201]. In a second study focused on the rice–striped rice stemborer pathosystem [202], 2 of the 13 assessed *C. suppressalis* endogenous sRNAs thought to represent novel *C. suppressalis* miRNAs, were shown to provide rice plants with resistance to striped rice stemborer infection. Moreover, feeding experiments on material sampled from the two rice amiRNA transformant lines showed that when highly expressed in rice, the two *C. suppressalis* targeting amiRNAs interfered with *C. suppressalis* larval development to greatly limit the ability of striped rice stem borer to infect and cause disease symptom development in both rice transformant lines [202]. In the closely related moth species *Chilo polychrysus* (dark-headed rice stemborer), the insect-specific miRNA, miR-14, which was identified as a central regulatory factor of the exoskeleton moulting process, was introduced into rice by molecular manipulation [203]. The resulting transformant lines, which highly accumulated the introduced miRNA *Cpo*-miR-14, were shown to be resistant to dark-headed rice stem borer infection [203]. *Helicoverpa armigera* (cotton bollworm) is a significant pest of a wide range of agricultural species, such as cotton, soybean, chickpea, and sunflower. As such, the amiRNA approach has been demonstrated to be successful for use as an alternative strategy to the in planta expression of *Bacillus thuringiensis* (Bt) toxins for management of *H. armigera* infection of crops [204]. Moreover, Agrawal et al. [204] transformed *N. tabacum* plants with the precursor transcript of miR-24, an insect miRNA that targets the *Chitinase* gene for RNA silencing in *H. armigera*. Expression analyses showed that amiR-24 accumulated to high levels in amiR-24 transformant lines and that *H. armigera* larvae fed exclusively on a tobacco transformant line diet ceased to continue through their moulting process and eventually died [204]. The success of amiRNA technology to manage cotton bollworm was confirmed in a subsequent study in tomato [205]. Specifically, transformation of tomato with an amiRNA transgene termed, amiR-HaEcR, which targeted the *H. armigera* ecdysone receptor (HaEcR)-encoding gene and which was delivered in the *Arabidopsis* miR319a precursor transcript, was shown to be effective at negatively impacting both the development and overall survival rate of *H. armigera* [205].

Preliminary experimentation has identified the amiRNA approach as a potentially promising additional tool for use in molecular plant biology research to engineer resistance against a range of pathogens in select plant species. However, each of the studies reported above was performed in a plant species readily amenable to molecular manipulation, and were not performed in the primary target species of the plant pathogen. Frequently, the primary target species of a destructive plant pathogen is an economically important crop species that is either difficult to successfully manipulate at the molecular level or is completely recalcitrant to genetic manipulation. Therefore, the use of the SIGS approach overcomes the significant and long-running hurdle to offering an effective plant pathogen management strategy for agronomically important plant species. Furthermore, SIGS offers the added benefit of responding to plant pathogens in “real time”, with the RNA component of SIGS formulations being readily changeable to target the pathogenic species most prevalent in each specific growing season. Moreover, climate change is strongly influencing season-to-season variation, with such climate variation also influencing the microorganismal composition of each cultivation region. Therefore, SIGS would allow a grower to rapidly adapt to such growing season variation, helping to ensure that the grower can maintain their crop yields. In theory, amiRNA-directed SIGS could also be used by a grower not only to more effectively manage an altered plant pathogen landscape, but to also directly respond to climate change itself. For example, miRNAs that regulate crucial plant developmental processes, such as the transition from vegetative to reproductive development, are well documented (e.g., *Arabidopsis* miRNAs, miR156, and miR172). Therefore, synthetic versions of these endogenous miRNAs could be topically applied via SIGS to either promote or delay flowering in a cropping species to align the timing of this key determinant of plant crop yield to when the environmental conditions are most favourable. SIGS application of synthetic versions of such miRNAs could also limit the negative impact that other forms of abiotic stress have on plant development, such as topically applying amiRNAs that control the expression of plant genes which mediate the adaptive response to drought, salinity, altered temperature, and nutrient deficiencies. Therefore, the addition of PEVs and/or AEVs to SIGS formulations will likely further enhance the efficacy of this innovative alternative technology. Moreover, PEV/AEV supplementation of SIGS formulations would likely (1) protect the RNA molecules encapsulated in the vesicles from the surrounding environment to increase the duration which the encapsulated RNA remains “active” post-application, and (2) enhance cellular uptake of its molecular cargo by the plant tissues to which the PEVs/AEVs are applied due to these vesicles naturally performing this role in cross-kingdom communication. Figure 5A provides a schematic of the proposed workflow for the isolation of PEVs for use as vehicles to deliver amiRNA silencing signals in SIGS formulations to (1) provide protection to the plant against viral, bacterial, or fungal pathogens (Figure 5B), or (2) alter the timing of plant development to adapt to environmental change (Figure 5C).

## 6. Conclusions

Although initially viewed as a form of cellular debris, the isolation of PEVs from a diverse array of edible, medicinal, herbal, horticultural, and agricultural plant species has now firmly established their role as a fundamentally important conduit for the transport of molecular cargo to direct both cell-to-cell and cross-kingdom communication. The diversity of the molecular cargo harboured by PEVs, composed of species-specific formulations of metabolites, lipids and proteins, strongly suggests that PEVs mediate essential signalling and communication roles in (1) normal plant development, (2) adaptation to an altered growth environment, and (3) defence against invading pathogens. Nucleic acids also make a considerable contribution to the molecular payload housed by PEVs, with much of the research attention to date focused on the potential role mediated by PEVs in ferrying the miRNA class of small regulatory RNA to sites of pathogen infection.

Under normal growth conditions, the MVB biogenesis pathway makes the predominant contribution to the global PEV population of a plant. However, when the growth environment is altered by either abiotic or biotic stress, the EXPO biogenesis pathway appears to also make a significant contribution to PEV production. To account for the considerable diversity in PEV physical properties, the vacuolar and autophagy biogenesis models have also been proposed to potentially contribute to PEV production. Like the EXPO biogenesis pathway, these two hypothesised PEV biogenesis pathways have been proposed to be induced when a plant encounters abiotic or biotic stressors.

Although considerable evidence has now accumulated that demonstrates bidirectional exchange of miRNAs and other classes of small regulatory RNAs between host plants and invading pathogens, a considerably smaller number of these studies have documented a role for PEVs (or pathogen-derived EVs) in mediating the transfer of these molecular signals. However, a more expansive role for EVs in this process seems apparent. More specifically, in the host plant, PEVs would be selectively packaged with specific miRNAs for their targeted delivery to sites of pathogen infection. At such sites, PEVs would release their concentrated cargo of miRNA silencing signals to direct a highly localised molecular-based defensive response against the pathogen. Similarly, in the pathogen, host gene targeting miRNAs would be loaded into pathogen-generated EVs for their delivery to host cells. Pathogen-generated EVs could also promote the internalisation of their infection, promoting molecular cargo by host plant cells to enhance pathogenicity in order to overcome the host’s defence mechanisms for further infection and/or disease progression.

This proposed role for PEVs in miRNA trafficking, together with their inherent natural properties, identifies these vessels as an essential additive to RNA formulations, providing an innovative enhancement to molecular-based approaches for crop protection in future agribiotechnology. More specifically, it can be envisaged that PEVs are packaged with amiRNAs to form SIGS formulations which may further improve the efficacy of this recent technology development. The addition of PEVs to SIGS formulations would protect, and thus ensure, the prolonged stability of the RNA once it is applied to plant surfaces. PEV addition would also likely promote the cellular uptake of the topically applied RNA. Provision of stability, together with enhancing cellular uptake of topically applied RNAs, will enable further applications of PEV-encapsulated RNAs outside of pathogen defence in both agricultural and aquacultural settings.

## Figures and Tables

**Figure 1 genes-17-00052-f001:**
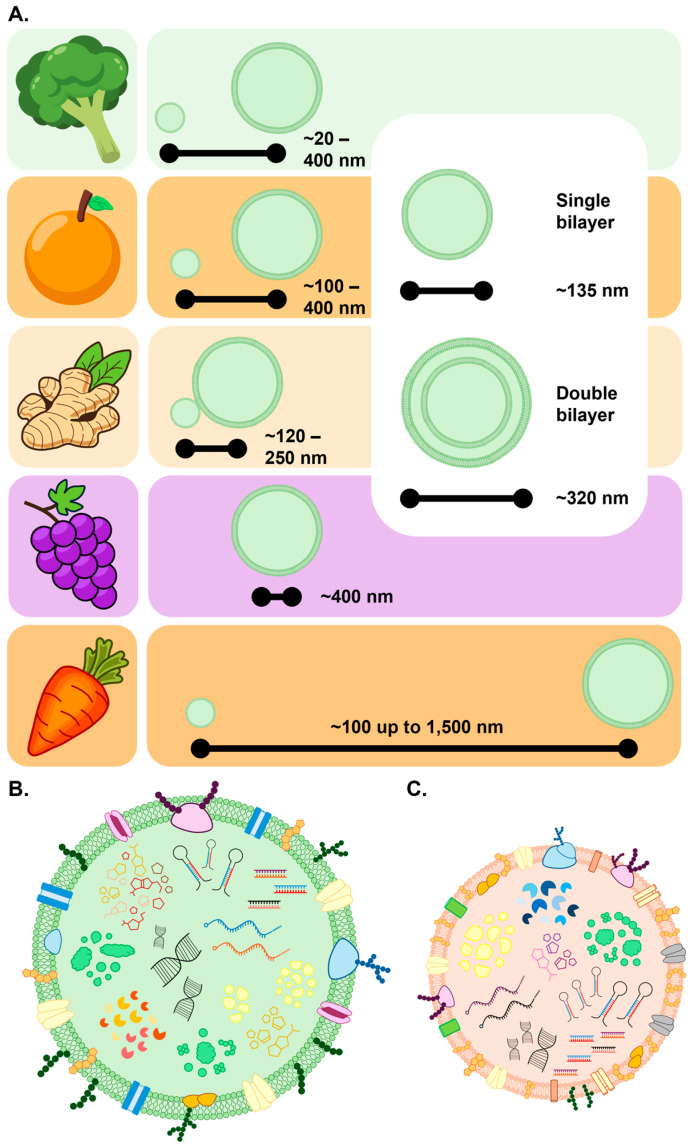
*Plant extracellular vesicle properties and characteristics.* (**A**) PEVs show considerable size variation, with the PEVs from broccoli, orange, ginger, grape, and carrot having average diameters of ~20–400 nm, ~100–400 nm, ~120–250 nm, ~400 nm, and ~100 up to ~1500 nm. Furthermore, single and double lipid bilayer PEVs also show average diameter differences of ~135 and ~320 nm, respectively. (**B**) The phospholipid bilayer which encapsulates a PEV is embedded with numerous protein classes, with each class performing an essential function, such as PEV cargo loading, targeted PEV transport, and ensuring PEV structural integrity. PEVs harbour an abundance of molecular cargo including proteins (green shaded shapes), enzymes (brown, pink, and orange shaded shapes), lipids (yellow shaded shapes), and metabolites (red, brown, and orange shaded shapes). In addition, PEVs harbour nucleic acids including DNA (black helixes), RNAs (e.g., mRNAs; blue and red squiggles), and sRNAs including both miRNAs (stem-loop shapes) and siRNAs (short ladders). (**C**) The phospholipid bilayer of animal EVs contains different protein classes that perform similar EV-related functions. Animal EVs also contain a complex mixture of proteins, lipids, enzymes, some metabolites, and a wealth of nucleic acids (e.g., DNA, mRNA, miRNA, and siRNA). However, like animal EV membrane protein composition, the internal molecular cargo profile of animal EVs is markedly distinct from that of PEVs.

**Figure 2 genes-17-00052-f002:**
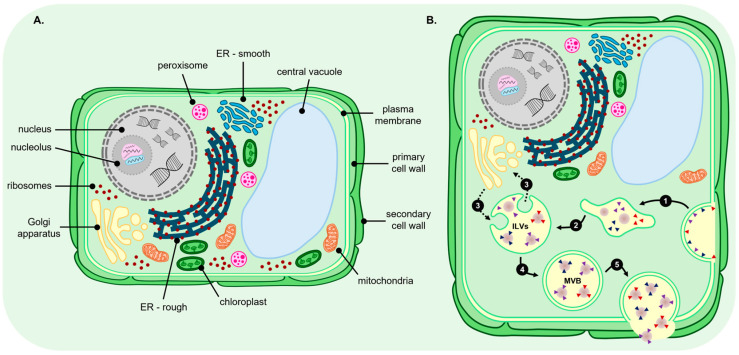
*Plant cell schematic and the primary PEV biogenesis pathway**.* (**A**) Plant cells are enclosed by a plasma membrane, and this is additionally surrounded by the primary cell wall. Other cell types or plant species also surround the plasma membraneof the plant cell with a secondary cell wall. Plant cells contain numerous large organelles, including the nucleus (which in turn harbours the nucleolus), smooth and rough endoplasmic reticulum, Golgi apparatus, and the central vacuole. Plant cells also harbour numerous other smaller-sized organelles such as chloroplasts, mitochondria, and peroxisomes, as well as an abundance of ribosomes in the cytoplasm. (**B**) *MVB biogenesis pathway:* (1) Likely identified by marker protein localisation (solid red, blue, and purple coloured triangles), endocytosis occurs at the plasma membrane to form an early endosome; (2) which via loading of specific cellular contents and molecular cargo transitions into a late endosome containing ILVs; (3) molecular cargo is exchanged bidirectionally between the late endosome and the Golgi apparatus to (4) result in MVB formation, and; (5) the marker proteins which identified the site of endocytosis may also direct MVB transport back to the plasma membrane for extracellular PEV release. The MVB biogenesis pathway likely forms the predominant pathway for the production of PEVs under normal growth conditions.

**Figure 3 genes-17-00052-f003:**
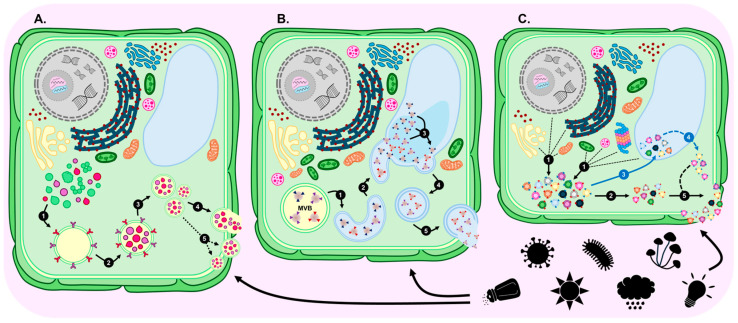
*Alternate PEV biogenesis pathways**.* (**A**) *EXPO biogenesis pathway:* (1) Rough ER and Golgi apparatus produce proteins lacking conventional signal peptides (pink and purple shaded shapes) which are loaded into exocyst-positive organelles (EXPOs). (2) EXPO loading is likely mediated by surface receptors and/or proteins and the loading of other molecular cargo results in (3) EXPO PEV production. The EXPO PEVs are transported to the plasma membrane where they (4) release their cargo extracellularly or (5) retain their molecular cargo and are released extracellularly as PEVs. (**B**) *Vacuolar biogenesis pathway (hypothesised):* (1) A small vacuole forms from the MVB and houses ILVs and/or PEVs. (2) The small vacuole fuses with the central vacuole and releases its harboured vesicles. (3) Likely dependent on the molecular cargo of each vesicle type, together with the surface marker protein profile, the vesicles are either retained by the central vacuole or loaded into a budding vacuolar PEV (either as a single vacuolar PEV or as multiple vesicles as per the original MVB (pictured)). (4) Post-loading, mature vacuolar PEVs are transported to the plasma membrane (5) for their extracellular release. (**C**) *Autophagy biogenesis pathway (hypothesised):* (1) Proteins or entire organelles (e.g., nucleus, Golgi apparatus, chloroplasts, mitochondria, smooth and rough ER, peroxisomes, and the proteosome (barrel of spheres)) can form targets of autophagosomes. (2 and 5) Likely due to protein composition, many autophagosome-derived PEVs will be directly transported to the plasma membrane for extracellular release. (3) Again, likely due to protein composition, some autophagosome-derived PEVs will initially be transported to the vacuole for processing. (4) Post-processing, specific autophagosome-derived PEVs will be transported back into the cytoplasm for (5) subsequent extracellular release. In contrast to the MVB biogenesis pathway, the documented EXPO biogenesis pathway and the hypothesised vacuolar and autophagosome biosynthesis pathways are likely activated by different forms of either biotic (viral, bacterial, and fungal pathogens) or abiotic (salinity, heat, flooding, and high light) stress as indicated by the black shaded cartoons.

**Figure 4 genes-17-00052-f004:**
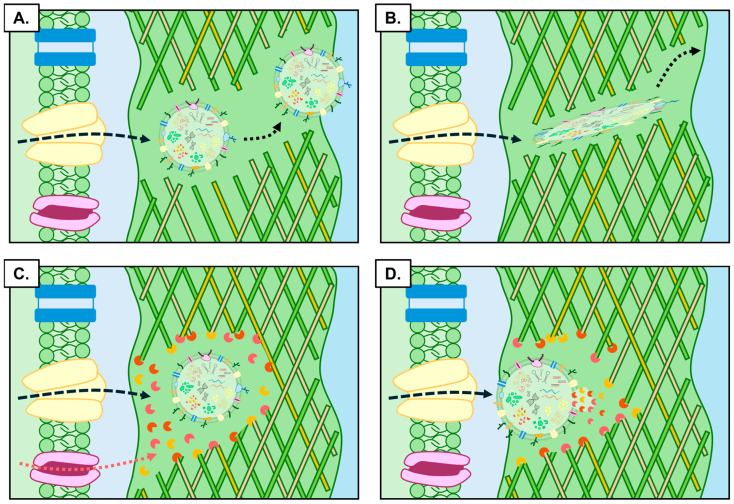
*Passive and activ**e mechanisms for PEVs to traverse the walls of plant cells*. (**A**) *Passive mechanism 1*: The inherent dynamic and viscoelastic properties of plant cell walls allow for transient rearrangements of structural components (i.e., cellulose microfibrils, glycans, pectin polysaccharides, and lignin) to facilitate PEV extracellular release. (**B**) *Passive mechanism 2*: Structural properties of the PEV itself may be modified to transform the spherical or cupular shape of a PEV into an elongated tubular structure to allow the PEV to passage through narrow pores of the plant cell wall. (**C**) *Active mechanism 1*: Cell-wall-modifying enzymes are released across the plasma membrane along with the PEVs which could locally modify cell wall composition to facilitate the construction of a transient pathway for PEVs to exist plant cells. (**D**) *Active mechanism 2*: PEVs themselves may harbour (internally) or express (external surface) the cell-wall-modifying enzymes required to reduce the integrity of the cell wall to allow for the PEV to traverse the walls which surround plant cells for their extracellular release.

**Figure 5 genes-17-00052-f005:**
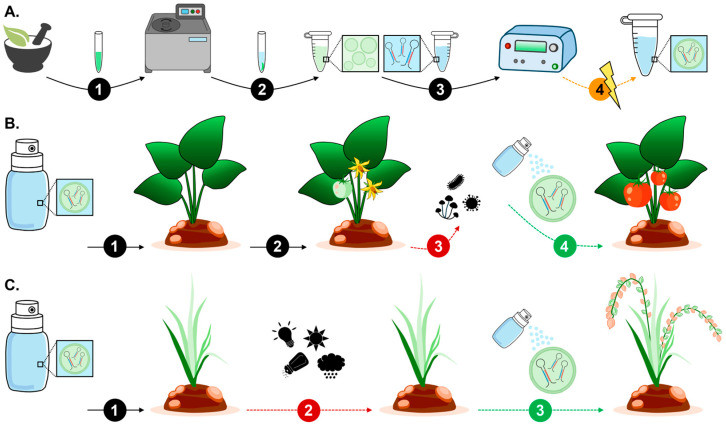
*PEV isolation, amiRNA loading, and the use of amiRNA-loaded PEVs in the horticultural and agricultural setting*. (**A**) Plant or algal samples are processed and added to an EV isolation buffer (1). (2) Ultracentrifugation is used to pellet the PEVs/AEVs out of the isolation buffer. (3) PEVs/AEVs resuspended in buffered solution which is then mixed with an in vitro synthesised amiRNA preparation. (4) Electroporation is used to promote amiRNA uptake by PEVs/AEVs. (**B**) *Application example 1—horticulture*: (1) amiRNAs can be designed to specifically target prevalent plant pathogens for a specific horticultural species, with the cultivated species surveyed throughout its vegetative (1) and reproductive (2) development. (3) If pathogen infection occurs (viral, bacterial, or fungal pathogen) at any stage of development, then (4) an amiRNA-loaded PEV/AEV formulation can be topically applied via SIGS to protect crop yield. (**C**) *Application example 2—agriculture*: (1) amiRNAs can be designed to specifically target key stress response genes. (2) Therefore, if abiotic stress is encountered during the growing season, the (3) amiRNA-loaded PEV/AEV formulation can be topically applied in real time via the SIGS approach to protect crop yield.

## Data Availability

Further information on the reviewed topic can be sourced via contacting the corresponding author.

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
