# Peer review of "Plant Extracellular Vesicles with Complex Molecular Cargo: A Cross-Kingdom Conduit for MicroRNA-Directed RNA Silencing"

_genes, 2026, doi:10.3390/genes17010052_

Round 1
Reviewer 1 Report
Comments and Suggestions for Authors
The topic of the manuscript is certainly relevant and it is clear that the authors have worked with a large amount of literature. Still, the overall presentation would benefit from clearer structure and more focused interpretation. The early sections dealing with the physical characteristics of PEVs are noticeably long and tend to repeat similar explanations. Several passages describe the same issues regarding vesicle size, morphology and isolation artefacts in only slightly different wording. This repetition makes it harder to maintain the line of argument and gives the impression that the text is not fully streamlined yet. I think these descriptions could be brought together and presented more concisely without losing any important information.
The section on biogenesis has a similar problem. The different secretion routes are introduced multiple times and the overlap between them makes the narrative somewhat difficult to follow. At the same time, some mechanisms are written in a way that suggests a level of certainty that is not yet supported by the field. Certain pathways remain hypothetical and should be presented as such. Making a clearer division between what is well established and what is still under debate would make this part more balanced. The microRNA-related sections include valuable information, but they rely mostly on description. Observations such as selective loading or the parallel existence of vesicle-dependent and vesicle-independent transport are interesting, yet the manuscript does not really explore what these findings mean in a broader biological context. The same applies to tiny RNAs: their origins are described very thoroughly, but the text does not discuss their possible functions or why they are abundant in vesicles. This leaves the reader with a list of observations but not a clear interpretation. Another point that affects clarity is the inconsistent use of terminology. Different labels for RNA species appear throughout the manuscript and are not always clearly distinguished. Given how diverse these RNA classes are, a more stable and transparent terminology would help the reader navigate the text more comfortably. Finally, some of the sentences are quite long and packed with clauses, which makes the manuscript heavier to read than it needs to be. Shortening and simplifying the structure of some sentences would greatly improve the readability.
Author Response
Response:
The authors thank Reviewer #1 for their positive and constructive assessment of our submitted article. We have thoroughly revised the text of the manuscript to address each of Reviewer #1’s comments. Further, below, we provide a specific response to each of Reviewer #1’s comments. Together with addressing the comments raised by Reviewer #2, the authors feel that addressing each Reviewer comment has improved the structure and readability of our article for its further consideration. Due to the extent of changes made to the original manuscript version we have submitted the revised manuscript version as a “clean copy” word document for ease of re-review.
Comment: The topic of the manuscript is certainly relevant and it is clear that the authors have worked with a large amount of literature. Still, the overall presentation would benefit from clearer structure and more focused interpretation. The early sections dealing with the physical characteristics of PEVs are noticeably long and tend to repeat similar explanations. Several passages describe the same issues regarding vesicle size, morphology and isolation artefacts in only slightly different wording. This repetition makes it harder to maintain the line of argument and gives the impression that the text is not fully streamlined yet. I think these descriptions could be brought together and presented more concisely without losing any important information.
Response: We thank the reviewer for identifying this issue. We have revised the text in the earlier sections of the review to make this section more streamlined and to remove repetitious statements. If any repetition does remain in the text, then it has been intentionally left in the revised version of the manuscript to provide tight linkage between individual sections of the review.
Comment: The section on biogenesis has a similar problem. The different secretion routes are introduced multiple times and the overlap between them makes the narrative somewhat difficult to follow. At the same time, some mechanisms are written in a way that suggests a level of certainty that is not yet supported by the field. Certain pathways remain hypothetical and should be presented as such. Making a clearer division between what is well established and what is still under debate would make this part more balanced.
Response: We thank the reviewer for outlining this issue. We have included text to more readily denote that the multivesicular body (MVB) and exocyst-positive organelle (EXPO) biogenesis pathways represent the predominant and somewhat characterised biogenesis pathways for extracellular vesicle production in plants. Similarly, additional text has been included in the revised version to outline that the vacuolar and autophagy biogenesis routes currently remain as hypothesised “alternative” biogenesis pathways for PEV production.
Comment: The microRNA-related sections include valuable information, but they rely mostly on description. Observations such as selective loading or the parallel existence of vesicle-dependent and vesicle-independent transport are interesting, yet the manuscript does not really explore what these findings mean in a broader biological context. The same applies to tiny RNAs: their origins are described very thoroughly, but the text does not discuss their possible functions or why they are abundant in vesicles. This leaves the reader with a list of observations but not a clear interpretation.
Response: We have modified the text in the revised manuscript version to attempt to address this reviewer concern. However, the majority of the examples discussed in our review remain in the early stages of experimental characterisation. Therefore, we have cautiously interpreted the limited amount of preliminary data available. Such an approach has also been taken here to ensure that we have not overstated or misinterpreted these early findings which cannot be supported by the data currently available.
The authors thank Reviewer #1 for identifying the issue surrounding our discussion on the more recently identified small RNA class, tiny RNAs. Due to the lack of currently available information on this small RNA class we have removed this section of text altogether in the revised manuscript version. Presently, there simply is not any additional information regarding tiny RNAs which would allow for us to further expand on our originally supplied interpretations.
Comment: Another point that affects clarity is the inconsistent use of terminology. Different labels for RNA species appear throughout the manuscript and are not always clearly distinguished. Given how diverse these RNA classes are, a more stable and transparent terminology would help the reader navigate the text more comfortably.
Response: We do thank Reviewer #1 for raising this concern. However, the authors are somewhat unsure on the specifics of this comment by Reviewer #1? We have checked the entire review to ensure consistency when discussing different RNA classes. However, in plants, microRNA is abbreviated to “miRNA” and a specific miRNA such as microRNA168 is abbreviated to “miR168”. We have, where warranted (if we believe confusion may arise), included the species prefix for many of the miRNAs which we discuss. For example, miR168 from Arabidopsis thaliana is presented as “Ath-miR168”.
However, it is important to note that the terminology for use when discussing fungal miRNAs, or other sRNA species from other plant pathogens is not currently as clearly defined as it is for plants. We have therefore used the most commonly adopted convention when discussing sRNAs from non-plant species, or we have used the naming system adopted by the authors whose work we are commenting on. As an example, we have used on numerous occasions throughout the manuscript’s text, the abbreviation “milRNA” for “miRNA-like small RNAs” from plant fungal pathogens as per the currently available literature. Similarly, the standard naming convention has been used when discussing insect miRNAs in the manuscript’s text.
All other discussed plant or animal sRNAs are defined on their first mention: on their first mention, the accepted abbreviation for the specific RNA class is provide in brackets, and therein, we use the abbreviated form. Specifically, abbreviations for microRNA, small RNA (or small regulatory RNA), small-interfering RNA, messenger RNA, primary-miRNA, precursor-miRNA, heterochromatin small-interfering RNA, trans-acting siRNA, phased siRNA, and microRNA-like RNAs.
Comment: Finally, some of the sentences are quite long and packed with clauses, which makes the manuscript heavier to read than it needs to be. Shortening and simplifying the structure of some sentences would greatly improve the readability.
Response: Thank you again Reviewer #1 for identifying this issue. All longer length sentences have been shortened and simplified in the revised manuscript version. Addressing this concern was highly helpful in improving both the structure and readability of the revised manuscript.
Reviewer 2 Report
Comments and Suggestions for Authors
Review of the manuscript
The manuscript describes the role of plant extracellular vesicles (PEVs) and their microRNA cargo in cellular communication, plant development, and plant responses to stress and pathogen attack.This is a very carefully prepared review article, extensive in scope and supported by several high-quality figures. The manuscript provides a comprehensive and well-structured overview of plant extracellular vesicles and their miRNA cargo, offering valuable insights into their roles in development, stress responses, and plant–pathogen interactions. But ms has some shortcomings. Please refer them for improvements to the manuscript.
Below, I point out the main remarks:
- Figure 2. - The figure provides a somewhat simplified description of the pathway, omitting several important aspects, including the detailed role of MVBs in cargo sorting, maturation, and vesicle trafficking. What role does the multivesicular body (MVB) biogenesis pathway play in the formation and release of plant extracellular vesicles (PEVs) under normal growth conditions?
- The text is generally well prepared; however, I noticed several instances where abbreviations are not defined upon their first use, as well as gene names should be written in italics.
- Throughout the text the authors frequently use permissive or tentative language such as may and can. I believe that many of these points are already well supported by current evidence, and the wording could be made more definitive to reflect established knowledge.
- It may be partly my own preference, given the large amount of information presented in the manuscript, but the authors could consider summarizing key elements in a table. This would make it easier to locate specific information in the primary literature and provide a clear, accessible overview of the core findings of the review.
- “Together, this series of elegant studies clearly show that amiRNA technology constitutes an innovative and effective plant pest management strategy.” The sentence should be formatted correctly.
- “In theory, amiRNA-directed SIGS could also be used by a grower to not only more effectively manage an altered plant pathogen landscape, but to directly respond to climate change itself.”I disagree with that statement. While amiRNA-directed SIGS (Spray-Induced Gene Silencing) is a promising tool for managing plant pathogens, there is currently no evidence that it can directly mitigate or respond to climate change. Its effects are limited to targeting specific genes in pathogens or pests, not altering environmental factors or climate-driven processes. And certainly, it is still very far from being a practical tool for growers. Current amiRNA-directed SIGS approaches remain largely experimental, and issues such as delivery efficiency, stability, regulatory approval, and cost would need to be fully addressed before field application.
- Conclusion - This section provides a clear and comprehensive overview of the emerging role of plant extracellular vesicles. However, while the text is informative, a few points could be improved for clarity and rigor: Some statements, such as the targeted delivery of miRNAs to pathogen infection sites, are presented somewhat definitively, even though experimental evidence is still limited. Modulating the language could better reflect the current stage of research. And in the end, the paragraph is very dense and could benefit from subdivision or the inclusion of a table summarizing the roles of PEV cargo and their functional outcomes. This would help readers quickly grasp the key points.
The authors should revise the entire manuscript considering these points. The subject is a worthy topic of investigation.
Author Response
Comments and Suggestions for Authors
Review of the manuscript
The manuscript describes the role of plant extracellular vesicles (PEVs) and their microRNA cargo in cellular communication, plant development, and plant responses to stress and pathogen attack.This is a very carefully prepared review article, extensive in scope and supported by several high-quality figures. The manuscript provides a comprehensive and well-structured overview of plant extracellular vesicles and their miRNA cargo, offering valuable insights into their roles in development, stress responses, and plant–pathogen interactions. But ms has some shortcomings. Please refer them for improvements to the manuscript.
Response: The authors thank Reviewer #2 for their positive and constructive assessment of our submitted article. We have thoroughly revised the text of the manuscript to address each of Reviewer #2’s comments. Further, below, we provide a specific response to each of Reviewer #2’s comments. Together with addressing the comments raised by Reviewer #1, the authors feel that addressing each Reviewer comment has greatly improved the structure and readability of our article for its further consideration. Due to the extent of changes made to the original manuscript version we have submitted the revised manuscript version as a “clean copy” word document for ease of re-review.
Below, I point out the main remarks:
- Figure 2. - The figure provides a somewhat simplified description of the pathway, omitting several important aspects, including the detailed role of MVBs in cargo sorting, maturation, and vesicle trafficking. What role does the multivesicular body (MVB) biogenesis pathway play in the formation and release of plant extracellular vesicles (PEVs) under normal growth conditions?
Response: Figure 2 is intentionally presented simplistically to provide a general overview of the MVB biogenesis pathway for PEV production. The accompanying text which describes Figure 2 provides more detail on the step-by-step processes of the pathway. Furthermore, presently, the individual steps of the MVB biogenesis pathway are more thoroughly documented in animals than they are in plants. In plants, it may remain too premature to provide a more mechanistically detailed model until further experimental characterisation of the pathway is reported.
To address the second half of this Reviewer #2 comment, we have altered the text of the revised manuscript version to state that the MVB biogenesis pathway makes the primary contribution to EV production in plants under normal growth conditions. We also supply additional text in the revised manuscript version which better highlights that the EXPO biogenesis pathway contributes more heavily to PEV production when the growth environment is altered (e.g., during environmental stress or upon pathogen infection).
In addition, in the revised manuscript we have made further wording changes to more accurately reflect that the vacuole and autophagy biogenesis pathways, at present, remain largely hypothetical. In the original submission these two biogenesis routes were presented as alternate PEV production pathways, so text changes were made to reflect that currently, these two pathways have only been proposed as different EV biogenesis sources in plants in the absence of any definitive experimental evidence.
- The text is generally well prepared; however, I noticed several instances where abbreviations are not defined upon their first use, as well as gene names should be written in italics.
Response: The authors thank Reviewer #2 for highlighting this issue. As part of preparing the revised manuscript we have carefully checked that all abbreviations are supplied on their first mention in the manuscript’s text. We have also checked to ensure that all gene names are presented in capital italics for plant genes and in lowercase italics for pathogen (fungus, bacteria, insect) genes as per the standards for these different systems. Similarly, where appropriate, we have presented transcript names capital italics for plant transcripts, and lower case italics for non-plant species. Upright capitals are used when discussing plant proteins and the first word of a non-plant protein name is capitalised.
- Throughout the text the authors frequently use permissive or tentative language such as may and can. I believe that many of these points are already well supported by current evidence, and the wording could be made more definitive to reflect established knowledge.
Response: We thank Reviewer #2 for identifying this issue. We have adjusted the text accordingly in the revised manuscript to provide a more definitive tone to our reporting on the current data.
- It may be partly my own preference, given the large amount of information presented in the manuscript, but the authors could consider summarizing key elements in a table. This would make it easier to locate specific information in the primary literature and provide a clear, accessible overview of the core findings of the review.
Response: The authors thank the reviewer for their suggestion of Table inclusion. As a review article, we would prefer to thoroughly summarise each key finding as text, rather than re-reporting the same data in table format. If this was a research article, then yes, the use of Tables to present key data would form an appropriate reporting style.
- “Together, this series of elegant studies clearly show that amiRNA technology constitutes an innovative and effective plant pest management strategy.” The sentence should be formatted correctly.
Response: We have replaced the identified sentence “Together, this series of elegant studies clearly show that amiRNA technology constitutes an innovative and effective plant pest management strategy” to “Preliminary experimentation has identified the amiRNA approach as a potentially promising additional tool for use in molecular plant biology research to engineer resistance against a range of pathogens in select plant species.” to address this reviewer comment.
- “In theory, amiRNA-directed SIGS could also be used by a grower to not only more effectively manage an altered plant pathogen landscape, but to directly respond to climate change itself.”I disagree with that statement. While amiRNA-directed SIGS (Spray-Induced Gene Silencing) is a promising tool for managing plant pathogens, there is currently no evidence that it can directly mitigate or respond to climate change. Its effects are limited to targeting specific genes in pathogens or pests, not altering environmental factors or climate-driven processes. And certainly, it is still very far from being a practical tool for growers. Current amiRNA-directed SIGS approaches remain largely experimental, and issues such as delivery efficiency, stability, regulatory approval, and cost would need to be fully addressed before field application.
Response: We outlined many of the issues raised by Reviewer #2 above which relate to the current challenges involved in the development of the SIGS approach and the use of topically applied RNAs in the original manuscript version. And yes, we do agree with the points raised above by Reviewer #2. Specifically, that currently, amiRNA technology remains as an experimental tool for plant molecular biology research, and is not at the stage of development where it could be rolled out on-scale for use in agriculture. However, based on our own unpublished experimental characterisation of amiRNA technology (unpublished), we remain of the opinion that in the future, amiRNA technology will form a tool which can be used to modulate plant development in order to “time” key developmental transitions in response to an altered environment. We have included this proposal as a future development for the expansion of the use of SIGS applications. Therefore, we respectfully disagree with the Reviewer’s comments regarding the potential future use of this promising technology.
- Conclusion - This section provides a clear and comprehensive overview of the emerging role of plant extracellular vesicles. However, while the text is informative, a few points could be improved for clarity and rigor: Some statements, such as the targeted delivery of miRNAs to pathogen infection sites, are presented somewhat definitively, even though experimental evidence is still limited. Modulating the language could better reflect the current stage of research. And in the end, the paragraph is very dense and could benefit from subdivision or the inclusion of a table summarizing the roles of PEV cargo and their functional outcomes. This would help readers quickly grasp the key points.
Response: The authors thank Reviewer #2 for this helpful suggestion regarding text improvement for the Conclusion. We have largely rewritten this section of the manuscript in the revised version (please see below) to address this Reviewer comment.
“Although initially viewed as a form of cellular debris, the isolation of PEVs from a diverse array of edible, medicinal, herbal, horticultural and agricultural plant species has now firmly established their role as a fundamentally important conduit for the transport of molecular cargo to direct both cell-to-cell and cross-kingdom communication. The diversity of the molecular cargo harbored by PEVs, composed of species-specific formulations of metabolites, lipids and proteins, strongly infers that PEV mediate essential signaling and communication roles in (1) normal plant development, (2) adaptation to an altered growth environment, and (3) defense against invading pathogens. Nucleic acids also make a considerable contribution to the molecular payload housed by PEVs with much of the research attention to date focused on the potential role mediated by PEVs in ferrying the miRNA class of small regulatory RNA to sites of pathogen infection.
Under normal growth conditions, the MVB biogenesis pathway makes the predominant contribution to the global PEV population of a plant. However, when the growth environment is altered by either abiotic or biotic stress, the EXPO biogenesis pathway appears to also make a significant contribution to PEV production. To account for the considerable diversity in PEV physical properties, the vacuolar and autophagy biogenesis models have also been proposed to potentially contribute to PEV production. Like the EXPO biogenesis pathway, these two hypothesized PEV biogenesis pathways have been proposed to be induced when a plant encounters abiotic or biotic stressors.
Although considerable evidence has now accumulated to demonstrate bidirectional exchange of miRNAs and other classes of small regulatory RNAs between host plants and invading pathogens, a considerably smaller number of these studies have documented a role for PEVs (or pathogen-derived EVs) in mediating the transfer of these molecular signals. However, a more expansive role for EVs in this process seems apparent. More specifically, in the host plant, PEVs would be selectively packaged with specific miRNAs for their targeted delivery to sites of pathogen infection. At such sites, PEVs would release their concentrated cargo of miRNA silencing signals to direct a highly localized molecular-based defensive response against the pathogen. Similarly, in the pathogen, host gene targeting miRNAs would be loaded into pathogen-generated EVs for their delivery to host cells. Pathogen-generated EVs could also promote the internalization of their infection promoting molecular cargo by host plant cells to enhance pathogenicity in order to overcome the host’s defense mechanisms for further infection and/or disease progression.
This proposed role for PEVs in miRNA trafficking, together with their inherent natural properties has identifies these vessels as an essential additive to RNA formulations, providing an innovative enhancement to molecular-based approaches for crop protection in future agribiotechnology. More specifically, it can be envisaged that PEVs are packaged with amiRNAs to form SIGS formulations which may further improve the efficacy of this recent technology development. The addition of PEVs to SIGS formulations would protect, and thus ensure, the prolonged stability of the RNA once it is applied to plant surfaces. PEV addition would also likely promote the cellular uptake of the topically applied RNA. Provision of stability, together with enhancing cellular uptake of topically applied RNAs, will enable further applications of PEV encapsulated RNAs outside of pathogen defense in both agricultural and aquacultural settings.”
The authors should revise the entire manuscript considering these points. The subject is a worthy topic of investigation.
Response: Thank you again Reviewer #2 for your thorough and insightful review, and for your helpful suggestions for manuscript improvement.
Round 2
Reviewer 1 Report
Comments and Suggestions for Authors
Upon review, I find the article to be accurate, well-written, and suitable for publication.